# Benchmark of plankton images classification: emphasizing features extraction over classifier complexity

Thelma Panaïotis[1,2], Emma Amblard[2,3], Guillaume Boniface-Chang[4], Gabriel Dulac-Arnold[5], Benjamin Woodward[6], Jean-Olivier Irisson[2]

[1]National Oceanography Centre, Southampton, UK
[2]Laboratoire d'Océanographie de Villefranche, Sorbonne Université, Villefranche-sur-Mer, France
[3]Fotonower, Paris
[4]Google Research, London
[5]Google Research, Paris
[6]CVision AI, Medford, MA, USA

*Correspondence to*: Thelma Panaïotis (thelma.panaiotis.pub@proton.me)

**Abstract.** Plankton imaging devices produce vast datasets, the processing of which can be largely accelerated through machine learning. This is a challenging task due to the diversity of plankton, the prevalence of non-biological classes, and the rarity of many classes. Most existing studies rely on small, unpublished datasets that often lack realism in size, class diversity and proportions. We therefore also lack a systematic, realistic benchmark of plankton image classification approaches. To address this gap, we leverage both existing and newly published, large, and realistic plankton imaging datasets from widely used instruments. We evaluate different classification approaches: a classical Random Forest classifier applied to handcrafted features, various Convolutional Neural Networks (CNN), and a combination of both. This work aims to provide reference datasets, baseline results, and insights to guide future endeavors in plankton image classification. Overall, CNN outperformed the classical approach but only significantly for uncommon classes. Larger CNN, which should provide richer features, did not perform better than small ones; and features of small ones could even be further compressed without affecting classification performance. Finally, we highlight that the nature of the classifier is of little importance compared to the content of the features. Our findings suggest that compact CNN (i.e. modest number of convolutional layers and consequently relatively few total parameters) are sufficient to extract relevant information to classify small grayscale plankton images. This has consequences for operational classification models, which can afford to be small and quick. On the other hand, this opens the possibility for further development of the imaging systems to provide larger and richer images.

## 1 Introduction

Plankton, defined as organisms unable to swim against currents, are crucial components of oceanic systems as they form the basis of food webs and contribute to organic carbon sequestration (Ware and Thomson 2005; Falkowski 2012). They have been the subject of scientific research for centuries (Péron and Lesueur 1810). The definition of planktonic organisms, based on motility and ecological niche rather than phylogeny, means that it encompasses a wide range of taxonomic clades

(Tappan and Loeblich 1973). Furthermore, within these clades, plankton is known to be particularly diverse (Hutchinson 1961). Thus, planktonic organisms cover a wide range of size (from a few micrometers to several meters), shape, opacity, color, etc. While some planktonic taxa are ubiquitous (e.g. copepods), many are rare and sparsely distributed (e.g. fish larvae, scyphomedusae) (Ser-Giacomi et al. 2018).

Historically, plankton was studied by sampling with nets and pumps followed by identification and counting by taxonomists. These approaches, still used today, are precise but time-demanding. Quantitative imaging and automated identification are now complementing traditional methods of plankton observation, with various imaging instruments developed to generate quantitative data (Lombard et al. 2019). Some of these instruments image collected samples, such as the ZooScan (Gorsky et al. 2010), the FlowCAM (Sieracki et al. 1998), or the ZooCAM (Colas et al. 2018). Others acquire images in situ, such as the Underwater Vision Profiler (UVP; Picheral et al. 2010, 2021), the In Situ Ichthyoplankton Imaging System (ISIIS; Cowen and Guigand 2008), the Imaging FlowCytobot (IFCB; Olson and Sosik 2007), or the ZooGlider (Ohman et al. 2019). These instruments vary significantly in terms of targeted size range, imaging technique, and deployment requirements, each necessitating distinct processing pipelines. Moreover, the growing availability and ease of use of these instruments are generating an ever-increasing volume of plankton imaging data. Most of this data is now processed through automated algorithms. Among the various processing tasks, detecting or identifying organisms is commonly performed using supervised machine learning, where an algorithm learns patterns from training data and then generalizes these patterns to new data. Despite significant advances in hardware for high-throughput plankton imaging, these new instruments do not always come with a solid and easy-to-use software pipeline (Bi et al., 2015 is a rare counter-example), leaving operators with the burden of coding or adapting one themselves. Even once the data is processed, many current analysis workflows still rely on aggregating and summarizing the classified images, since the usual statistical tools used in ecology are not meant to handle such large amounts of data points. This limits our ability to leverage the full richness of these new datasets (Malde et al. 2020).

Automated classification of plankton images is a challenging computer science task. To begin with, planktonic communities (Ser-Giacomi et al., 2018), and therefore the resulting image datasets (Eftekhari et al., 2025; Schröder et al., 2019), exhibit significant class imbalance. In other words, a few classes contribute to a substantial part of the dataset, while others classes are poorly represented. This specificity of plankton image datasets contrasts with standard benchmark image datasets where classes are almost evenly distributed: between 732 and 1300 images for each of the 1000 classes in ImageNet (Russakovsky et al. 2015). As a consequence, rare planktonic classes are usually harder to predict for automated algorithms (Lee et al. 2016; Van Horn and Perona 2017; Schröder et al. 2019), although classes with highly distinctive morphologies could still be correctly classified even with few training images (Kraft et al., 2022). Secondly, planktonic organisms encompass a wide range of taxa and form a morphologically heterogeneous group, varying in size, shape and opacity. More specifically, certain classes can exhibit significant intraclass variation: for instance, when morphological differences arise from life stages (e.g.,

doliolids) or when a class includes diverse, but rare, objects grouped together, as they are too uncommon to warrant separate classes (e.g., fish larvae). This variability can lead to confusion between classes (Grosjean et al. 2004). In addition to diverse classes of living organisms, real-world plankton image datasets comprise a considerable amount of non-living objects, such as marine snow aggregates or bubbles (Benfield et al. 2007); these classes often constitute the majority of the datasets (Ellen et al. 2019; Schröder et al. 2019; Irisson et al. 2022). Finally, plankton images collected by quantitative instruments are typically low in resolution (with edges measuring only a few hundred pixels or less) and are often grayscale or with little variation in color; therefore the distinction among classes needs to be made from a relatively small amount of information.

Historically, the automatic classification of plankton images involved training machine learning classifiers using handcrafted features extracted from the images. These manually extracted features – intended to capture plankton traits (observable characteristics, primarily morphological) − aim to summarize the image content in numerical form, providing a concise representation that facilitates the classification process. Typical handcrafted features were global image moments (size, average gray, etc.; Tang et al. 1998), texture features such as gray-level co-occurrence matrices (Hu and Davis 2005), or shape features from Fourier transforms of the contour (Tang et al 1998). Classifiers included Support Vector Machines (SVM; Luo et al. 2004; Hu and Davis 2005; Sosik and Olson 2007), Random Forests (RF; Gorsky et al. 2010) or Multi-Layer Perceptrons (MLP; Culverhouse et al. 1996). Several studies compared various classifiers trained on a common set of features, revealing varying results depending on the dataset, but ultimately no significant difference in their performance (Grosjean et al. 2004; Blaschko et al. 2005; Gorsky et al. 2010; Ellen et al. 2015, 2019). This suggests that the performance of classical approaches is not driven by the classifier as much as by the number and diversity of features that are fed to it. Indeed, classification performance usually increases with a richer set of features (Blaschko et al. 2005). Nevertheless, this may not be true if some features are redundant or introduce noise into the data, hence the importance of feature selection (Sosik and Olson 2007; Guo et al. 2021b). Because handcrafted features are designed for a particular imaging system, a single universal set that works across all instruments is difficult to define; the optimal set of features tends to be instrument and dataset dependent (Orenstein et al. 2022). One solution would be to define a very large, universal feature set and leave it to the classifier to select the relevant ones for each task. But this would be a challenging task, as it requires both expertise in biology, for many taxa (to know what to extract), and in computer science (to know how to do it); feature engineering has therefore emerged as a complete research field (Guyon and Elisseeff 2003). In the following, we will refer to these two-step methods (1 − handcrafted feature extraction and 2 − classification) as "classic approaches", in contrast to the "deep approaches" introduced later, which bypass manual feature design by training feature extractors that automatically learn relevant features for the task at hand (Irisson et al., 2022).

Among classifiers, RF is a tree-based ensemble learning method that has shown high accuracy and versatility among computer vision tasks (Hastie et al. 2009). Each decision tree in the "forest" is trained on a random subset of the data (i.e. bootstrap), and at each step, it considers a random selection of predictors (or features) to split the data according to labeled

classes. The tree keeps splitting until it reaches a stopping point, such as a maximum number of splits. During prediction, each object passes through the tree until it reaches a terminal leaf, where it is classified based on the majority class within that leaf. By averaging the results from multiple trees, RF reduces the risk of overfitting (Breiman 2001). Fernández-Delgado et al. 2014, who evaluated the performances of nearly 180 classifiers on various datasets, concluded that RF outperformed all others. Gorsky et al. 2010 previously reached this conclusion on a ZooScan images dataset, resulting in a widespread use of RF classifiers later on. The IFCB data processing pipeline also switched from SVM to RF (Anglès et al. 2015). Finally, EcoTaxa (Picheral et al. 2017), a web application dedicated to the taxonomic annotation of images, initially implemented a RF classifier to classify unlabeled images.

However, since 2015, an increasing proportion of plankton image classification studies have employed deep learning methods, especially Convolutional Neural Networks (CNN). CNN are a kind of artificial neural network, typically used for pattern recognition tasks like image segmentation or classification. Their architecture is inspired from the visual cortex of animals, where each neuron reacts to stimuli from a restricted region (Dyck et al. 2021). In the case of an image classification task, a CNN directly takes an image as input (as opposed to classic approaches for which image features need to be extracted first), transforms it in various ways (the "Convolutional" part), combines the resulting features as input for a set of interconnected "neurons" that further reduce the information (the "Neural Network" part), and finally outputs a probability for the image to belong to each class; the class of highest probability is chosen as the predicted label. In contrast to classical approaches described above, the classification task with CNN is performed in a single step, where the feature extractor and the classifier are trained simultaneously. This process optimizes the deep features specifically for the classification task. Moreover, those features can be used to train any kind of classifier, often resulting in better classification performance than with handcrafted features (Orenstein and Beijbom 2017).

CNN, first developed in 1990 (LeCun et al. 1990) and popularized in 2012 (Krizhevsky et al. 2012), were applied to plankton image classification for the first time in 2015, during a challenge hosted on the online platform Kaggle[1]. Since then, numerous studies have demonstrated the effectiveness of CNN in recognising plankton images (Dai et al. 2016; Lee et al. 2016; Luo et al. 2018; Cheng et al. 2019; Ellen et al. 2019; Lumini and Nanni 2019; Schmid et al. 2020). On a few plankton images datasets, CNN have proven to reach higher prediction accuracy than the classical approach of handcrafted features extraction followed by classification (Orenstein et al. 2015; Kyathanahally et al. 2021; Irisson et al. 2022). Currently, research on the classification of plankton images, or images of any other type of marine organisms, is dominated by CNN (Irisson et al. 2022; Rubbens et al. 2023, Eerola et al., 2024). While CNN remain a dominant method for image classification, they have been surpassed by vision transformers (Vaswani et al. 2017), a newer state-of-the-art approach. However, vision transformers are less data-efficient than CNN, requiring larger datasets and greater computational resources

---

[1] https://www.kaggle.com/c/datasciencebowl/

for effective training (Raghu et al. 2021). When applied to plankton image classification, vision transformers have shown only marginal improvements over CNN (Kyathanahally et al. 2022; Maracani et al. 2023).

A relatively recent review (Irisson et al. 2022) revealed that over 175 papers have addressed the topic of automated classification of plankton images. As shown earlier, a few compared classifiers explicitly, with varying outcomes. But overall, these 100+ studies used different datasets, often only one per study, and most of which were not publicly released. The datasets varied in terms of number of classes and number of images, two factors that significantly affect performance.

They also reported different performance metrics and the one most commonly reported (global accuracy) is unrepresentative for unbalanced datasets (Soda 2011). Indeed, out of the 10 most cited papers in the field (Irisson et al. 2022), 8 conducted a plankton classification experiment, but only 4 reported per class metrics or a confusion matrix (others only report global metrics such as accuracy). A similar pattern is observed among the papers cited here: of the 33 papers that performed a plankton classification task, only half reported metrics beyond global metrics (Table S1). Looking at the bigger picture, it

appears that performance has remained relatively stable over time, while the taxonomic classification tasks became increasingly difficult since, with richer and larger datasets, more taxa were labeled (Irisson et al. 2022). This suggests that classifiers improved, although this is unquantifiable for all the reasons above. Earlier plankton image datasets were modest in size, typically containing a dozen or a few dozen of classes (Benfield et al., 2007), but were crucial for establishing the first classification methods. Building on that foundation, three major plankton image datasets have been published and used

in several studies (Table 1), while a few other studies have focused on smaller versions of these datasets (Dai et al. 2016; Zheng et al. 2017; Lumini and Nanni 2019). These benchmark datasets share several important characteristics: they are large (though this is debatable for PlanktonSet 1.0), representative of true data (with minimal alteration of class distribution and inclusion of all classes, such as detritus or miscellaneous), and accessible online. This highlights that a move towards standardization and intercompatibility is ongoing. Beyond publishing large reference datasets, as we strive to do in this work,

another avenur for progress is the collection of many diverse, albeit smaller, datasets. This is typically the first step for the creation of "universal" foundation-type models. The push towards more open and reproducible science has helped in this respect and several local datasets have been published: e.g. Table 1 in Kareinen et al. (2025), Table 2 in Eerola et al. (2024).

**Table 1: Common plankton images benchmark datasets.**

| Name | References | Imaging instrument | Composition | | Relevant publications |
|------|------------|--------------------|-------|--------|------------------------|
| | | | Images | Classes | |
| WHOI-plankton | Orenstein et al. 2015; Sosik, Peacock, and Brownlee 2015 | IFCB | 3.5 M | 103 | Callejas et al., 2025; Ciranni et al., 2025; Lee et al. 2016; Dai et al. 2017; Orenstein and Beijbom 2017; Cui et al. 2018; Hassan et al., 2025; Kraft et al., 2022; Kyathanahally et al. 2021, 2022; Langeland Teigen et al., 2020; Liu et al., 2018; Maracani et al. 2023; Venkataramanan et al., 2021 |
| ZooScanNet | Elineau et al. 2024 | ZooScan | 1.4 M | 93 | Callejas et al., 2025; Ciranni et al., 2025; Guo and Guan, 2021; Malde and Kim 2019; Schröder et al. 2019; Kyathanahally et al. 2021, 2022; Maracani et al. 2023 |
| PlanktonSet 1.0 | Cowen et al. 2015 | ISIIS | 30,336 | 121 | Dieleman et al. 2016; Du et al., 2020; Geraldes et al., 2019; Guo and Guan, 2021; Guo et al., 2021a; Langeland Teigen et al., 2020; Li and Cui, 2016; Li et al., 2019; Py et al. 2016; Rodrigues et al. 2018; Uchida et al. 2018; Kyathanahally et al. 2021, 2022; Langeland Teigen et al., 2020; Maracani et al.; Yan et al., 2017 |

Currently, despite several years of active research on the topic and while CNN have been applied to plankton images for more than five years (Luo et al. 2018), a systematic, global comparison of classifier performance is still lacking. Leveraging both previously published and new published plankton imaging datasets, the motivation for this study is to provide such a systematic, operational benchmark that evaluates practical and accessible approaches suitable for real-world applications. This includes starting with a classical feature-based image classification approach and exploring a few deep-learning methods. All are applied on large, realistic, and publicly released datasets from six commonly used plankton imaging instruments, to encompass some of the variability in imaging modalities, processing pipelines, and target size ranges present in plankton imaging. For the classical approach, we use the handcrafted features natively extracted by the software associated with the instrument, assuming that they were engineered to be relevant for those images, and a RF classifier,

given its popularity and performance on plankton images. For the deep approach, our base model is a relatively small and easy to train CNN (MobileNet V2), readily accessible to non ML specialists and below state of the art hardware. In addition to this benchmark, we perform additional comparisons to tackle the following questions: (i) In which conditions do CNN strongly improve classification performance over the classical approach? (ii) Is per-class weighting of errors effective to counter the effect of class imbalance in plankton datasets? (iii) How rich do features need to be for plankton images classification: are larger CNN needed or, on the contrary, can features be compressed? (iv) What are the relative effect of features (deep vs. handcrafted) and classifier (MLP vs. RF) on classification performance?

## 2 Material and method

### 2.1 Datasets

#### 2.1.1 Imaging tools

We used datasets from six widely used plankton imaging instruments, each with distinct properties such as deployment methods or the size range of targeted organisms (Table 2). For a detailed review of these instruments, refer to Lombard et al. 2019.

Table 2: Main characteristics of the plankton imaging instruments used to collect the datasets.

| Instrument | Deployment | Covered size range | Reference |
|---|---|---|---|
| FlowCAM | Ex situ (laboratory, ship) | 20 to 200 µm | (Sieracki et al. 1998) |
| IFCB | In situ (mooring) | 10 to 100 µm | (Olson and Sosik 2007) |
| ISIIS | In situ (ship-towed) | < 1 mm to several cm | (Cowen and Guigand 2008) |
| UVP6 | In situ (CTD rosette, mooring, AUV) | 620 µm to a few cm | (Picheral et al. 2021) |
| ZooCAM | Ex situ (laboratory, ship) | > 300 µm | (Colas et al. 2018) |
| ZooScan | Ex situ (laboratory) | 200 µm to a few cm | (Gorsky et al. 2010) |

#### 2.1.2 Image processing

Each imaging tool had its own specific image processing and feature extraction pipeline. The motivation here is to use these tools "out of the box", as other plankton ecologists would. ISIIS data was processed using Apeep (Panaïotis et al. 2022), and features were extracted using Scikit-image (Walt et al. 2014). The IFCB data processing relied on several MATLAB scripts (Sosik and Olson 2007) to segment objects and extract different types of features. The UVPapp application (Picheral et al. 2021) was developed to process UVP6 images and extract features. Both ZooScan and FlowCAM data were processed using

ZooProcess (Gorsky et al. 2010), which generates crops of individual objects together with a set of features, extracted by ImageJ (Schneider et al. 2012). The processing of ZooCam data was very similar to the processing of ZooScan and FlowCAM data (Colas et al. 2018). Thus, for all datasets, each grayscale image was associated with a set of handcrafted features, which depended on the instrument but were mostly global features, related to shape and gray-levels, and a label.

### 2.1.3 Datasets assembling and composition

All datasets were generated in a similar way: complete, real-world datasets were sorted by human operators; All classifications were reviewed by one independant operator for each dataset. Except for IFCB and ZooCAM, samples particularly rich in some rare classes were added to the dataset (all images, not just those of the class of interest). Classes still containing fewer than ~100 objects were merged into a taxonomically and/or morphologically neighboring class. If no relevant merging class could be found, objects were assigned to a miscellaneous class together with objects impossible to classify. Therefore, every single object from the original samples was included in the classification task, ensuring that the metrics computed on these datasets were as relevant to a real-world situation as possible. The IFCB images were taken from Sosik et al. 2015 (years 2011-2014); the images for other instruments were taken from EcoTaxa (Picheral et al. 2017), with the permission of their owners. Full references for each dataset are provided in Table 3. The number of images in the resulting datasets ranged from 301,247 to 1,592,196, in 32 to 120 classes (Table 3). As expected, the datasets collected in situ (ISIIS, UVP6, and IFCB) were particularly rich in marine snow and other non-living objects, resulting in a low proportion of plankton.

To assess performance at a coarser taxonomic level, which may be sufficient in some applications and is more comparable to older papers tackling automated classification of plankton images (e.g. Culverhouse et al. 1996; Sosik and Olson 2007; Gorsky et al. 2010), each class was assigned to a broader group (Tables 4, S2-S7). Each class/group was then categorized as planktonic or non-planktonic (i.e. detritus and imaging artifacts), allowing metrics to be computed for planktonic organisms only, excluding the, sometimes dominant, non-living objects (Table 3). The datasets were split, per class, into 70% for training, 15% for validation and 15% for testing, once, before all experiments. This split ensured that the majority of the data was used for training, maximizing model learning, while preserving a sufficient portion for validation and testing (at least 10 objects for the rarest classes in FlowCAM and ISIIS datasets).

**Table 3: References and dataset composition in terms of the numbers of images, classes and handcrafted features, as well as the proportion of plankton (i.e. living organisms, as opposed to detritus and imaging artifacts).**

| Instrument | Dataset reference | Composition | | | |
|---|---|---|---|---|---|
| | | # images [min; max per class] | Classes | Features | % plankton |

| FlowCAM | (Jalabert et al. 2024) | 301,247 [74 ; 69,085] | 93 | 47 | 36.2 |
|---|---|---|---|---|---|
| ISIIS | (Panaïotis et al. 2024) | 408,166 [70 ; 321,335] | 32 | 31 | 15.3 |
| UVP6 | (Picheral et al. 2024) | 634,459 [87 ; 508,817] | 54 | 62 | 7.7 |
| ZooCAM | (Romagnan et al. 2024) | 1,286,590 [81 ; 204,132] | 93 | 48 | 67.8 |
| ZooScan | (Elineau et al. 2024) | 1,451,745 [90 ; 241,731] | 120 | 48 | 71.2 |
| IFCB | (Sosik et al. 2015) | 1,592,196 [90 ; 1,177,499] | 69 | 72 | 12.6 |

## 2.2 Classification models

Each dataset was classified using different models, described below. The training procedure was the same for all models and datasets: (i) models were fitted to the training split, according to a loss metric, (ii) hyperparameters were assessed based on the same loss metric but computed on the independent validation split to limit overfitting, (iii) the model with optimal hyperparameters was used to predict the never-seen-before test split, only once, and various performance metrics were computed.

The RF classifiers were implemented using Scikit-learn (Pedregosa et al. 2011). The CNN models were implemented using Tensorflow (Abadi et al. 2016). Training and evaluation were performed on two Linux machines, depending on the model: a Dell server equipped with a Quadro RTX 8000 GPU and a node of the Jean-Zay supercomputer, equipped with a V100 SXM2 GPU.

The code to reproduce all results is available at https://doi.org/10.5281/zenodo.17937437 (Panaïotis and Amblard 2025).

### 2.2.1 Classic approach

A RF classifier was trained on handcrafted features extracted from images by the software dedicated to each instrument. Their number ranged from 31 to 72 depending on the software (Table 3). Most features were global features, computed on the whole object: morphological features were computed on the object silhouette; gray-levels features were summaries of the distribution of gray levels in the object. In the case of IFCB, additional texture features were extracted, in the form of gray level co-occurrence matrices. The diversity of features is known to be crucial for the performance of the classifiers (Blaschko et al. 2005).

The loss metric used during training and validation was categorical cross-entropy, which optimizes the model's confidence in predicting the correct class by minimizing the difference between predicted probabilities and actual labels. While this helps

improve accuracy, it does not directly optimize for accuracy itself, which is based solely on whether predictions are correct, not on the confidence of those predictions. In terms of hyperparameters, the number of features used to compute each split was set to the square root of the number of features (which is the default for a classification task, Hastie et al. 2009) and the minimum number of samples in a terminal node was set to 5. The optimal number of trees was investigated using a grid search procedure, over the values 100, 200, 350, and 500; for each, the classifier was fitted on the training split and evaluated on the validation split. The number of trees leading to the lowest validation loss was selected. This classic approach is illustrated in the first row of Fig. 1.

### 2.2.2 Convolutional neural network

Since our goal here is to assess the performance of easy-to-use, turnkey models that most research teams should be able to deploy, we chose a rather small CNN (MobileNet V2; Sandler et al. 2019), as our reference model. In addition, we also evaluated the performance of much larger CNN: EfficientNet V2 (Tan and Le 2021), in its S and XL versions.

Images were resized and padded to match the input dimensions required by each CNN model (MobileNet V2: 224×224×3; EfficientNet V2 S: 384×384×3; EfficientNet V2 XL: 512×512×3). Since each image was originally single-channel, the single channel was replicated across the typical three color channels used in CNN. To preserve aspect ratio, each image was resized so that its longest side equaled the model's input size, then padded to a square format using the median value of the border pixels to maintain a homogeneous background (Orenstein et al. 2015). Since all images are resized and padded to a common pixel grid, the large natural size variation of plankton is compressed, limiting the amount of scale-specific detail that can be exploited by the CNN. Finally, the grayscale channel was replicated to create three identical channels and achieve the desired shape. Since training a CNN from scratch is time and data-consuming, we applied transfer learning by using a feature extractor pre-trained on the ImageNet dataset. The pre-trained feature extractor could be used as it is, as the features extracted by a model trained on generic datasets have also proven to be relevant for other tasks (Yosinski et al. 2014), such as plankton classification (Orenstein and Beijbom 2017; Rodrigues et al. 2018; Kyathanahally et al. 2021). But they can also be fine-tuned on the target dataset to achieve better performance (Yosinski et al. 2014), which is what we did here, for each dataset.

In a CNN, the typical classifier following the feature extractor is a MLP. To prevent overfitting, we added a dropout layer (rate = 0.5) immediately after the feature vector, preventing the model from relying on a few key neurons only (Srivastava et al. 2014) This was followed by a fully connected layer with either 600 or 50, depending on the model, to explore how the layer size impacts performance. Finally, the model ended with a classification head, the size of which depended on the number of classes to predict. This resulted in 4.5 M parameters for the smaller CNN and 208 M for the larger one. All models are described in Fig. 1.

Data augmentation (Shorten and Khoshgoftaar 2019) was used to improve model generalization ability and performance, especially for rare classes. Images from the training set were randomly flipped vertically and horizontally, zoomed in and out (up to 20%), and sheared (up to 15°). Such a process increases the diversity of examples seen during training, improving generalization ability of the model (Dai et al. 2016). Images were not rotated because objects from a few classes had a specific orientation (e.g. vertical lines in the ISIIS dataset, or some organisms that have a specific orientation in datasets collected in situ). As for the RF, the loss metric was the categorical cross entropy. At the end of each training epoch (i.e. a complete run over all images in the training split), both loss and accuracy were computed on the validation split, to check for overfitting, and model parameters were saved.

The feature extractor, fully connected and classification layers were trained for 10 epochs (5 epochs for EfficientNets). Monitoring the loss on the validation set revealed that this was sufficient for exhaustive training (Fig. S1). The optimizer used the Adam algorithm, with a decaying learning rate from an initial value of 0.0005 and a decay rate of 0.97 per epoch. Similarly to the optimization of the number of trees of the RF models, the number of training epochs was optimized by retaining the parameters associated with the epoch presenting the minimum validation loss, hence reducing overfitting (Smith 2018).

### 2.2.3 Hybrid approaches

Finally, to discriminate the effect of the feature extractor (either handcrafted or deep) and the classifier (either a RF or a MLP), the deep features produced by the fine-tuned MobileNet V2 (n = 1792) were used to train a RF classifier. Furthermore, to compare RF trained on similar numbers of features and to evaluate the importance of feature richness, we reduce the dimension of those deep features from 1792 to 50 using a principal component analysis (PCA) fitted on the training set only, before feeding them into the RF classifier. These two "hybrid" approaches are illustrated in the last two rows of Fig. 1.

### 2.2.4 Class weights

In an unbalanced dataset, well-represented classes are given more importance because examples from these classes are more frequent in the loss calculation, while very small classes are almost negligible. As a result, performance on these small classes is often very poor (Luo et al. 2018; Schröder et al. 2019). To address this imbalance, training data can be resampled to achieve a more balanced distribution (e.g. oversampling poorly represented classes and/or undersampling dominant classes), a set of methods known as dataset-level approaches (Sun et al. 2009). Alternatively, the classifier can be tuned so that the misclassification cost is higher for small classes (i.e. algorithm-level approaches). Although both types of methods were shown to improve classification performance in some situations (e.g. for a binary classification task, McCarthy et al. 2005), resampling forces the model to learn on an artificial, balanced class distribution; when the real-world data have a different (often skewed) distribution, the learned decision thresholds become mis-calibrated and performance degrades

(Moreno-Torres et al. 2012; González et al. 2017). Thus, a class-weighted loss was implemented to increase the cost of misclassifying rare plankton classes. Class weights can be set as the inverse frequency of classes, or smoother alternative such as root or fourth-root of the inverse frequency (Cui et al. 2019), which gives, for class $i$:

$$w_i = \left(\frac{\max(c)}{c_i}\right)^{0.25}$$

The effect of these per-class weights was investigated by training both weighted and non-weighted versions of a RF on native features and of the reference CNN (Mob + MLP$_{600}$; Fig. 1).

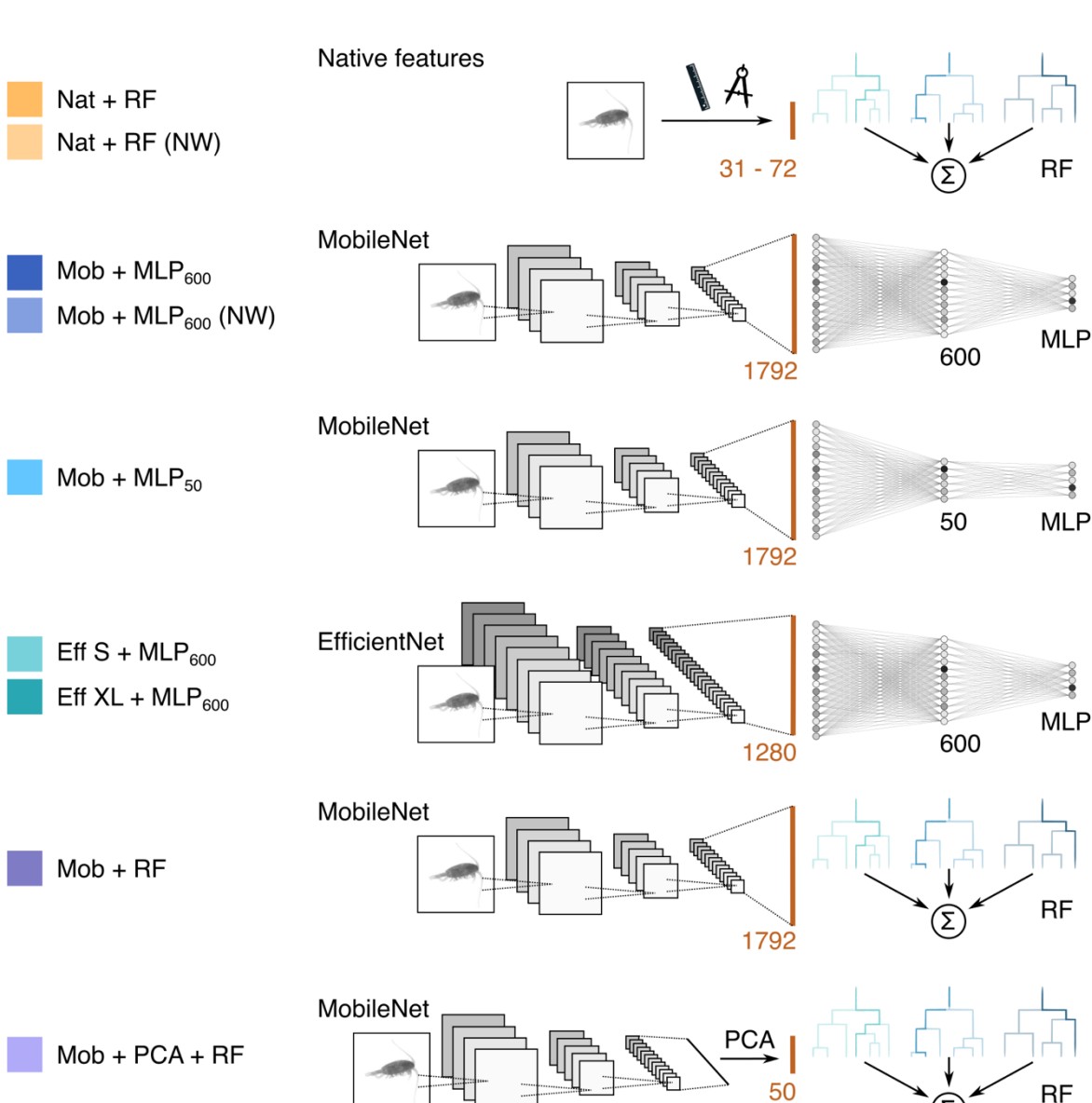

Figure 1: Description of the models tested. Each model consists of a feature extractor and a classifier, and is named accordingly. For each model, the brown line represents the feature vector and its length is indicated. For MLPs, the number in subscript gives the size of the fully connected layer. RF = Random Forest, MLP = Multilayer Perceptron, NW = no weights (i.e. learning not weighted by class size), PCA = Principal Component Analysis. The colors defined here are consistent with other figures. The MobileNet V2 with a fully connected layer of size 600 (Mob + MLP$_{600}$, in dark blue) will be considered as a reference model and repeated in all figures.

### 2.2.5 Model evaluation

After each model in Fig. 1 was trained and tuned for either the number of trees (for classical models) or the number of epochs (for CNN) on each dataset, models were evaluated on the test split, to which they had not been previously exposed. Usual metrics were computed: accuracy score (percentage of objects correctly classified), balanced accuracy, macro-averaged F1-score, micro-averaged F1-score, class-wise precision (percentage correct in the predicted class) and recall (percentage correct within the true class).

In datasets with strong class imbalance − such as many plankton datasets − accuracy alone can be misleading. For instance, in an 11-class dataset with one dominant class comprising 90% of the data (and each of the other classes making up only 1%), a classifier that always predicts the dominant class would achieve 90% accuracy but would provide no insight into the ten minority classes. A random classifier that draws labels according to the empirical class distribution would yield a lower-bound 81% accuracy ($0.9^2 + 10 \times 0.01^2$). This baseline reflects the underlying distribution while still producing a full confusion matrix that can be used to compute metrics such as precision and recall. In addition, the balanced accuracy score, computed as the simple average of per-class recall scores, was also computed, as it is a better estimate of model performance in such a scenario (Kelleher et al. 2020).

Furthermore, in the case of plankton datasets, the dominant classes are often not plankton (detritus, mix, etc.). The accuracy value is mostly driven by these classes (Orenstein et al. 2015) and, therefore, does not provide any information about the performance on plankton classes, which are often the subject of study. To focus on these classes, we also computed the average of precision and recall per class, weighted by the number of objects in the class, but using only plankton classes, i.e. the target classes (Owen et al., 2025). Averaged plankton recall gives a direct indication of the proportion of planktonic organisms that were correctly predicted, while averaged plankton precision reflects how free the predicted plankton classes are from false positives.

## 3 Results

### 3.1 Training time

Training and evaluation times were always shorter for the classical approach (using pre-extracted handcrafted features and a RF classifier) than for CNN (which combined feature extraction and classification). Running on 12 CPU cores, gridsearch, training, and evaluation for the RF classifier based on native features took less than an hour for the smallest dataset (ISIIS, ~400,000 objects) and a few hours for the IFCB dataset (~1.6 M objects). The extraction of handcrafted features could not be reliably timed, as it is performed using very different software, but is usually in the order of hours for about a million

objects. In contrast, it took 5h to train the MobileNet V2 + MLP$_{600}$ for 10 epochs on the ISIIS dataset but 15h for the same number of epochs on the IFCB dataset, using a Quadro RTX 8000 GPU.

## 3.2 Benchmark performance of MobileNetV2, our reference model

On the six large and realistic plankton image datasets included in this study, a small CNN model (MobileNetV2) trained with per-class weights achieved strong performance while remaining easy to implement. The balanced accuracy across all classes ranged from 79% to 90%, with plankton class precision and recall reaching 80%, except for ISIIS and UVP6 datasets. These benchmark results are further compared to other approaches in the following sections.

**Table 4: Classification report for detailed classes in the ZooScan dataset. Reported values are F1-scores. N test indicates the number of objects in the test set for each class. A colored version of this table is available in the Supplementary Materials (Table S7).**

| Class | Grouped | N test | Nat + RF | Mob + MLP600 | Eff S + MLP600 | Mob + PCA + RF |
|-------|---------|--------|----------|--------------|----------------|----------------|
| *Plankton classes* | | | | | | |
| Actinopterygii | Actinopterygii | 289 | 23.8 | 87.9 | 91.6 | 94.5 |
| egg<Actinopterygii | Actinopterygii | 689 | 35.3 | 88.3 | 88.3 | 90.5 |
| Neoceratium | Alveolata | 53 | 0.0 | 92.3 | 89.5 | 92.7 |
| Noctiluca | Alveolata | 980 | 54.6 | 92.7 | 90.2 | 92.5 |
| Amphipoda | Amphipoda | 125 | 0.0 | 82.7 | 86.1 | 90.1 |
| Cumacea | Amphipoda | 78 | 30.4 | 91.2 | 94.0 | 94.8 |
| Hyperiidea | Amphipoda | 289 | 26.1 | 90.2 | 93.4 | 92.8 |
| Annelida | Annelida | 349 | 21.3 | 85.0 | 85.9 | 87.5 |
| larvae<Annelida | Annelida | 50 | 0.0 | 72.9 | 75.2 | 75.0 |
| part<Annelida | Annelida | 149 | 35.7 | 86.2 | 85.4 | 88.2 |
| Tomopteridae | Annelida | 83 | 7.0 | 92.1 | 91.8 | 89.6 |
| Fritillariidae | Appendicularia | 1820 | 28.1 | 89.7 | 88.9 | 90.5 |
| Oikopleuridae | Appendicularia | 4967 | 39.4 | 94.2 | 94.5 | 95.0 |
| tail<Appendicularia | Appendicularia | 1243 | 48.6 | 85.2 | 84.4 | 86.9 |
| trunk | Appendicularia | 193 | 0.0 | 67.3 | 67.1 | 72.4 |
| Chaetognatha | Chaetognatha | 7859 | 75.4 | 97.3 | 97.6 | 97.9 |
| head<Chaetognatha | Chaetognatha | 190 | 0.0 | 56.9 | 69.8 | 72.4 |
| tail<Chaetognatha | Chaetognatha | 555 | 15.3 | 73.0 | 75.0 | 77.6 |
| cirrus | Cirripedia | 60 | 9.1 | 68.5 | 59.5 | 68.6 |
| cypris | Cirripedia | 147 | 0.0 | 87.9 | 92.8 | 91.8 |
| nauplii<Cirripedia | Cirripedia | 649 | 0.0 | 92.2 | 92.4 | 94.3 |
| Evadne | Cladocera | 5003 | 17.1 | 96.8 | 97.1 | 97.4 |
| Penilia | Cladocera | 3592 | 39.9 | 96.8 | 97.0 | 97.7 |

| | | | | | | |
|---|---|---|---|---|---|---|
| Podon | Cladocera | 292 | 0.0 | 88.3 | 87.8 | 87.6 |
| Acartiidae | Copepoda | 8853 | 24.2 | 95.5 | 95.4 | 95.9 |
| Calanidae | Copepoda | 6190 | 33.0 | 96.3 | 96.4 | 97.0 |
| Calanoida | Copepoda | 22713 | 57.6 | 94.3 | 94.3 | 94.9 |
| Calocalanus pavo | Copepoda | 71 | 2.7 | 84.2 | 85.5 | 89.9 |
| Candaciidae | Copepoda | 1767 | 11.9 | 95.5 | 95.1 | 95.5 |
| Centropagidae | Copepoda | 6890 | 32.8 | 94.6 | 94.6 | 95.1 |
| Copilia | Copepoda | 99 | 0.0 | 88.5 | 94.2 | 95.1 |
| Corycaeidae | Copepoda | 3576 | 28.5 | 96.3 | 96.6 | 97.2 |
| Eucalanidae | Copepoda | 183 | 16.8 | 88.4 | 90.2 | 91.3 |
| Euchaetidae | Copepoda | 1019 | 21.3 | 94.2 | 94.1 | 96.2 |
| Haloptilus | Copepoda | 407 | 31.8 | 95.6 | 95.4 | 96.5 |
| Harpacticoida | Copepoda | 832 | 0.2 | 90.7 | 92.7 | 93.1 |
| Heterorhabdidae | Copepoda | 355 | 0.0 | 87.6 | 86.2 | 89.3 |
| Metridinidae | Copepoda | 2439 | 14.7 | 94.6 | 94.6 | 95.7 |
| Oithonidae | Copepoda | 9847 | 59.2 | 96.6 | 96.6 | 97.0 |
| Oncaeidae | Copepoda | 3070 | 9.1 | 93.4 | 94.2 | 94.8 |
| Pontellidae | Copepoda | 1080 | 54.8 | 97.0 | 96.5 | 98.6 |
| Rhincalanidae | Copepoda | 35 | 52.0 | 70.2 | 78.3 | 85.3 |
| Sapphirinidae | Copepoda | 162 | 0.0 | 91.8 | 91.2 | 91.9 |
| Temoridae | Copepoda | 4549 | 23.4 | 96.0 | 96.0 | 96.9 |
| Ctenophora | Ctenophora | 137 | 0.0 | 67.0 | 72.3 | 81.1 |
| cyphonaute | cyphonaute | 1334 | 29.8 | 98.4 | 98.5 | 98.4 |
| larvae<Luciferidae | Decapoda | 98 | 16.4 | 95.2 | 95.4 | 97.9 |
| larvae<Porcellanidae | Decapoda | 748 | 64.2 | 96.2 | 97.4 | 98.3 |
| megalopa | Decapoda | 213 | 27.9 | 95.9 | 95.2 | 96.7 |
| protozoea<Penaeidae | Decapoda | 59 | 0.0 | 84.2 | 87.6 | 92.3 |
| protozoea<Sergestidae | Decapoda | 89 | 0.0 | 78.5 | 71.7 | 81.0 |
| zoea<Brachyura | Decapoda | 1750 | 40.0 | 95.7 | 96.7 | 97.5 |
| zoea<Galatheidae | Decapoda | 759 | 1.3 | 88.1 | 88.3 | 89.3 |
| Doliolida | Doliolida | 1461 | 37.7 | 93.2 | 92.4 | 93.8 |
| larvae<Echinodermata | Echinodermata | 76 | 0.0 | 80.6 | 76.6 | 84.0 |
| pluteus<Echinoidea | Echinodermata | 361 | 26.8 | 86.7 | 87.8 | 89.7 |
| pluteus<Ophiuroidea | Echinodermata | 542 | 13.4 | 91.0 | 92.5 | 92.0 |
| Eumalacostraca | Eumalacostraca | 3453 | 61.3 | 91.4 | 91.7 | 92.4 |
| Eumalacostraca potentially protozoea | Eumalacostraca | 225 | 26.1 | 83.0 | 81.4 | 83.8 |
| larvae<Mysida | Eumalacostraca | 14 | 0.0 | 72.7 | 88.9 | 82.8 |
| Mysida | Eumalacostraca | 120 | 76.5 | 86.4 | 91.6 | 94.4 |
| Harosa | Harosa | 244 | 1.6 | 76.7 | 75.1 | 74.2 |
| Isopoda | Isopoda | 83 | 67.1 | 98.8 | 97.6 | 98.2 |
| Atlanta | Mollusca | 68 | 0.0 | 84.8 | 83.9 | 90.9 |

| | | | | | | |
|---|---|---|---|---|---|---|
| Bivalvia<Mollusca | Mollusca | 777 | 12.6 | 95.0 | 95.5 | 95.8 |
| Cavolinia inflexa | Mollusca | 662 | 58.2 | 97.5 | 96.2 | 97.2 |
| Creseidae | Mollusca | 767 | 47.4 | 93.7 | 94.0 | 94.2 |
| Creseis acicula | Mollusca | 1294 | 67.6 | 94.5 | 94.4 | 94.9 |
| Cymbulia peroni | Mollusca | 14 | 0.0 | 80.0 | 72.7 | 76.5 |
| egg<Mollusca | Mollusca | 129 | 1.5 | 76.7 | 77.0 | 75.7 |
| Gymnosomata | Mollusca | 79 | 60.4 | 92.8 | 95.7 | 95.6 |
| Limacinidae | Mollusca | 2113 | 25.3 | 96.1 | 96.3 | 96.9 |
| part<Mollusca | Mollusca | 255 | 2.2 | 61.9 | 55.3 | 60.9 |
| Actiniaria | other_Cnidaria | 22 | 16.7 | 93.0 | 93.3 | 89.8 |
| ephyra | other_Cnidaria | 179 | 36.7 | 86.4 | 91.5 | 91.3 |
| Hydrozoa | other_Cnidaria | 579 | 13.6 | 74.6 | 75.1 | 78.4 |
| Obelia | other_Cnidaria | 147 | 18.2 | 85.9 | 85.7 | 88.5 |
| part<Cnidaria | other_Cnidaria | 125 | 0.0 | 14.8 | 44.0 | 44.6 |
| calyptopsis | other_Crustacea | 1205 | 12.2 | 93.5 | 94.3 | 93.3 |
| larvae<Stomatopoda | other_Crustacea | 245 | 46.5 | 95.6 | 96.5 | 98.4 |
| metanauplii<Crustacea | other_Crustacea | 37 | 0.0 | 81.8 | 85.3 | 93.7 |
| nauplii<Crustacea | other_Crustacea | 845 | 4.6 | 91.5 | 91.8 | 93.3 |
| Ostracoda | other_Crustacea | 1169 | 46.4 | 96.4 | 96.7 | 97.6 |
| part<Crustacea | other_Crustacea | 3065 | 2.6 | 63.2 | 65.3 | 68.2 |
| Pyrosomatida | Pyrosomatida | 75 | 22.2 | 93.9 | 95.4 | 94.8 |
| Foraminifera | Rhizaria | 469 | 25.7 | 89.7 | 89.8 | 90.4 |
| Phaeodaria | Rhizaria | 8106 | 55.1 | 96.6 | 96.2 | 96.7 |
| endostyle | Salpida | 135 | 16.0 | 60.4 | 58.2 | 61.4 |
| juvenile<Salpida | Salpida | 67 | 0.0 | 82.3 | 84.0 | 81.9 |
| nucleus | Salpida | 222 | 11.5 | 68.6 | 71.4 | 74.7 |
| Salpida | Salpida | 2460 | 42.1 | 92.9 | 92.3 | 93.4 |
| Bassia | Siphonophorae | 15 | 0.0 | 57.1 | 50.0 | 56.0 |
| bract<Abylopsis tetragona | Siphonophorae | 185 | 34.9 | 91.2 | 89.0 | 89.9 |
| bract<Diphyidae | Siphonophorae | 2185 | 12.0 | 85.9 | 86.0 | 87.9 |
| eudoxie<Abylopsis tetragona | Siphonophorae | 98 | 0.0 | 90.3 | 92.1 | 89.6 |
| eudoxie<Diphyidae | Siphonophorae | 525 | 2.9 | 84.3 | 86.9 | 89.9 |
| gonophore<Abylopsis tetragona | Siphonophorae | 199 | 12.1 | 90.9 | 90.2 | 93.5 |
| gonophore<Diphyidae | Siphonophorae | 2460 | 30.0 | 93.2 | 93.4 | 94.2 |
| nectophore<Abylopsis tetragona | Siphonophorae | 173 | 20.7 | 88.6 | 87.6 | 91.7 |
| nectophore<Diphyidae | Siphonophorae | 4417 | 63.1 | 92.9 | 92.2 | 93.1 |
| nectophore<Hippopodiidae | Siphonophorae | 17 | 18.2 | 73.3 | 81.1 | 85.7 |
| nectophore<Physonectae | Siphonophorae | 1386 | 59.5 | 87.4 | 81.8 | 84.7 |
| part<Siphonophorae | Siphonophorae | 412 | 0.0 | 66.8 | 67.4 | 69.5 |
| Physonectae | Siphonophorae | 16 | 0.0 | 43.5 | 48.5 | 66.7 |
| siphonula | Siphonophorae | 144 | 19.2 | 90.3 | 86.1 | 89.0 |

| | | | | | | |
|---|---|---|---|---|---|---|
| Coscinodiscus | Stramenopiles | 1075 | 41.2 | 97.3 | 96.8 | 97.2 |
| actinula<Solmundella bitentaculata | Trachylina | 19 | 0.0 | 68.8 | 78.9 | 82.4 |
| Aglaura | Trachylina | 455 | 57.9 | 91.8 | 91.7 | 93.0 |
| Liriope<Geryoniidae | Trachylina | 34 | 0.0 | 52.0 | 73.0 | 78.7 |
| Rhopalonema velatum | Trachylina | 373 | 49.1 | 85.6 | 85.2 | 87.2 |
| Solmundella bitentaculata | Trachylina | 56 | 3.5 | 67.4 | 70.6 | 73.4 |
| *average* | | | *22.9* | *85.5* | *86.6* | *88.5* |
| *Non plankton classes* | | | | | | |
| artefact | artefact | 7718 | 76.7 | 80.8 | 80.0 | 79.8 |
| badfocus<artefact | badfocus | 6046 | 19.6 | 63.1 | 62.9 | 63.1 |
| bubble | bubble | 2432 | 19.0 | 92.2 | 91.0 | 91.2 |
| detritus | detritus | 36260 | 55.2 | 82.9 | 81.4 | 81.6 |
| fiber<detritus | fiber | 6708 | 62.9 | 74.6 | 74.7 | 74.8 |
| Insecta | Insecta | 169 | 27.1 | 84.3 | 86.9 | 89.6 |
| egg<other | other_egg | 2015 | 59.7 | 92.2 | 91.0 | 92.4 |
| other<living | other_living | 40 | 16.3 | 39.2 | 59.3 | 73.7 |
| seaweed | seaweed | 1272 | 35.3 | 68.0 | 68.2 | 66.3 |
| *average* | | | *41.3* | *75.2* | *77.3* | *79.2* |

## 3.3 Rare classes are where CNN outperform classical approaches

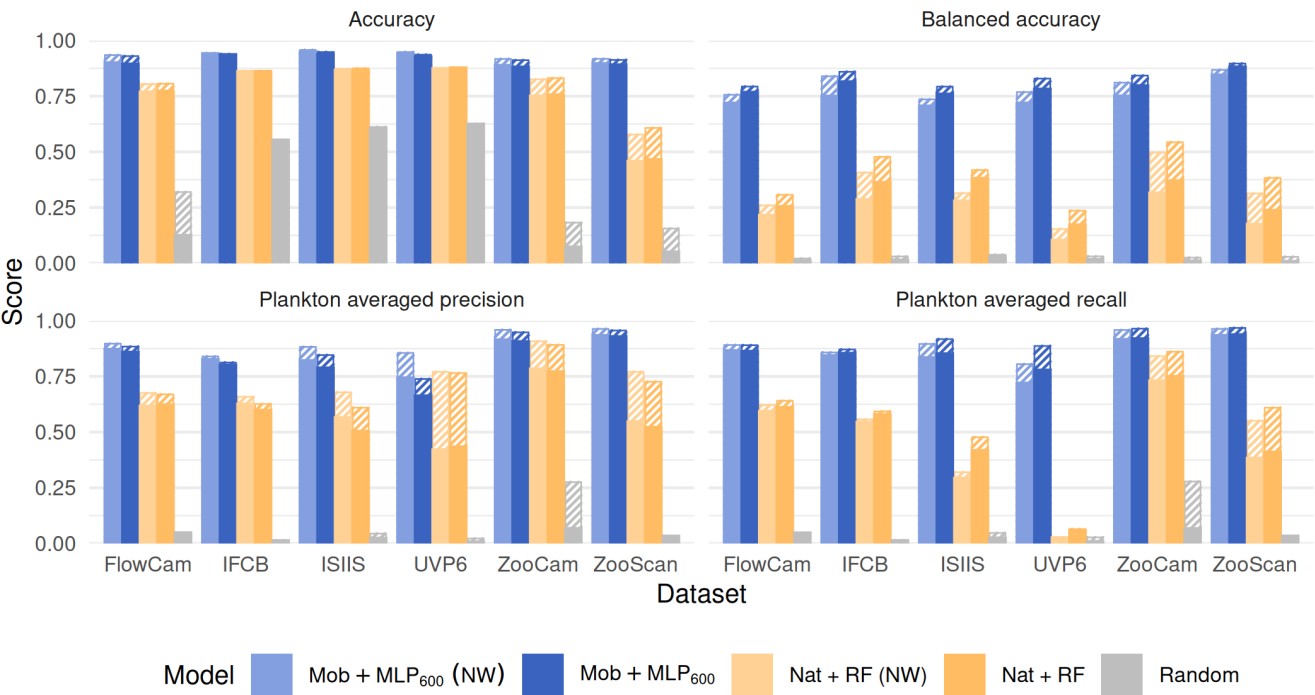

**Figure 2: Performance comparison between a small CNN (Mob + MLP600), a RF trained on handcrafted features and a random classifier on all six datasets. Both class weighted and non-weighted versions of the models were evaluated. The models are described in Fig. 1. Plain bars show the value of each metric at the finest taxonomic level, striped bars show the value after regrouping objects into broader ecological groups. All values, including F1-scores, are reported in Table S8.**

In terms of overall accuracy, the CNN only showed a modest improvement on five datasets compared with the classical approach of using handcrafted features and an RF classifier (+3.5% to +13.8%) (Fig. 2). The exception was the UVP6 dataset, where the improvement was more pronounced (> 40%) The use of class weights slightly decreased the accuracy of both the deep and classical approaches, as it focused training on small classes and less on large classes, which account for more in the computation of accuracy. Note that a random classifier achieved 55%, 61% and 63% accuracy on the detritus-dominated IFCB, ISIIS and UVP6 datasets, respectively. While the accuracies of all non-random models were higher, they must be gauged in terms of the increase over the random model and not in absolute terms.

Deep approaches showed much higher balanced accuracies than classical ones, as well as improved precisions and recalls averaged over plankton classes; this was true both with and without weights (Fig. 2). The balanced accuracy of the random classifier was very poor in all datasets, confirming that this metric is more relevant in datasets with many small classes. The same applies for F1-scores: macro-F1 captures the failure of the random classifiers, while micro-F1 mirrors accuracy (Fig.

S2). The improvements brought by CNN were associated with the fact that they performed better on non-dominant classes (e.g. Tables 4, S2-S7).

Class weights improved balanced accuracy for both deep (up to +8.2% for the UVP6 dataset) and classical approaches (up to +18.0% for the UVP6 dataset). Thus, as expected, giving more weight to small classes improved their learning by the classifier, but this was especially true for RF models. Weighting decreased plankton precision for both models, on all datasets: errors involving samples from large classes were less penalized, resulting in a greater contamination of plankton classes, i.e. lower precision. Symmetrically, the use of class weights improved the recall of plankton classes for all models

(except MobileNet on the FlowCam dataset). Again, this improvement is expected since plankton classes, which typically contain fewer images than non-plankton ones (e.g. detritus), are given more weight, reducing the number of false negatives, i.e. increasing recall. Since applying class weights improved detection of underrepresented classes (primarily plankton), only the weighted versions of each model will be evaluated in the subsequent analysis.

### 3.4 Small CNN are sufficient for plankton image classification

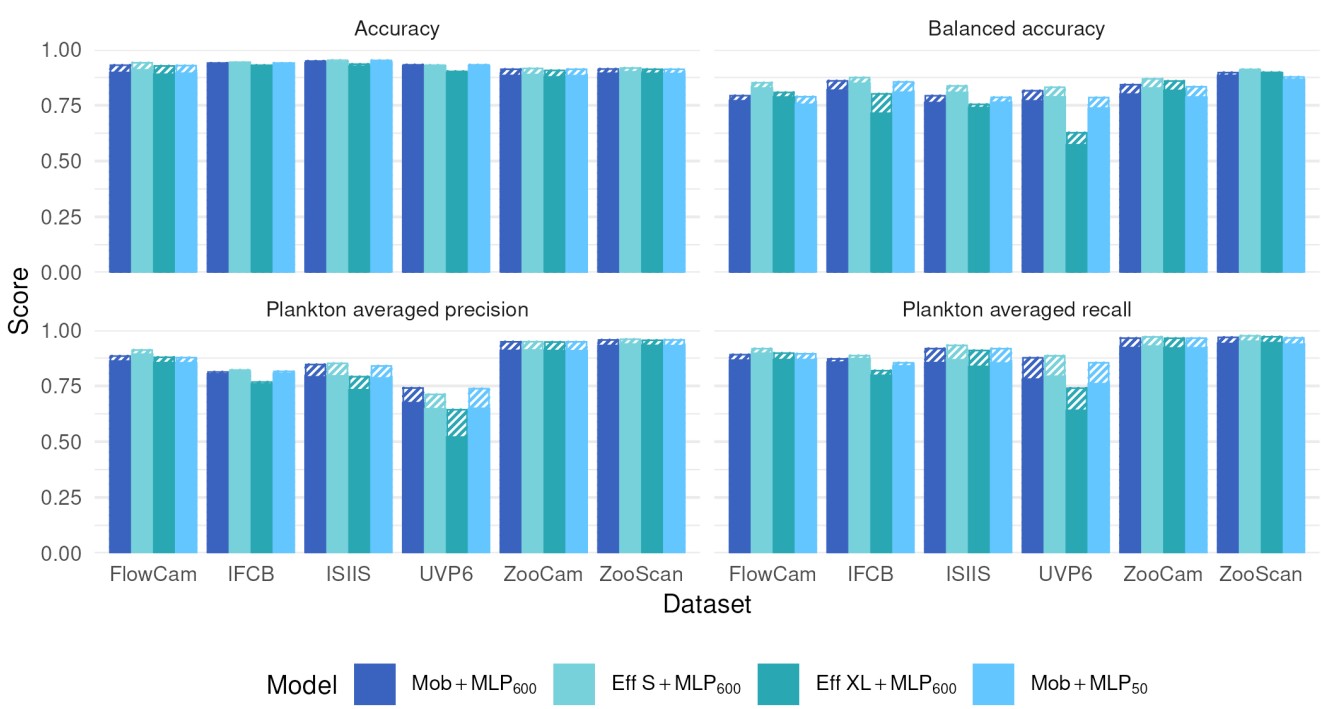

**Figure 3: Performance comparison between our reference CNN (Mob + MLP600), a CNN with a larger feature extractor (Eff S + MLP600 and Eff XL + MLP600) and a MobileNet followed by a smaller MLP (Mob + MLP50) on all six datasets. The models are described in Fig. 1. Plain bars show the value of each metric at the finest taxonomic level, striped bars show the value after regrouping objects into broader ecological groups. All values, including F1-scores, are reported in Table S8.**

Using a larger and supposedly richer feature extractor, such as EfficientNet S or EfficientNet XL, did not markedly improve performance metrics (Fig. 3). If anything, performance was lower with EfficientNet XL, likely due to immediate overfitting after the first epoch, causing the model to adhere too closely to the training data and impair its ability to generalize. This may be due to the relatively small training dataset, which, in proportion to the number of parameters in the model, increases the risk of overfitting. The effect was especially pronounced with the UVP6 dataset, which is not only small (~635,000 images)

but also has a low proportion of plankton images (7.7%); both balanced accuracy and plankton-specific metrics (average precision and recall) were notably impacted. On the other hand, compressing the features before classification, by using a fully connected layer of size 50 instead of 600 after the MobileNet feature extractor, did not reduce classification performance (Fig. 3). Both results suggest that a relatively small model is enough to extract all informative content from the small, grayscale plankton images in these datasets.

**3.5 The features are more important than the classifier**

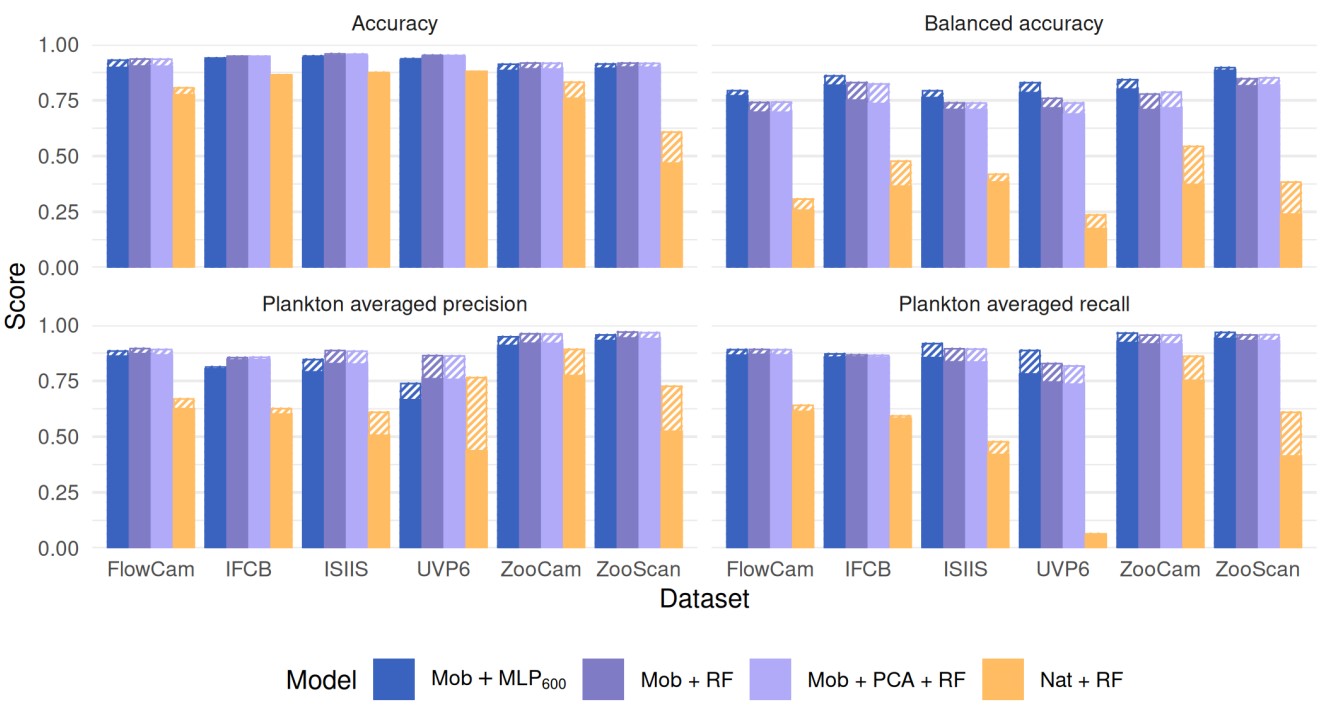

**Figure 4: Performance comparison between our reference CNN (Mob + MLP600), a RF trained on deep features extracted by a MobileNet V2 without (Mob + RF) and with (Mob + PCA + RF) feature reduction, and a RF trained on handcrafted features on all six datasets. The models are described in Fig. 1. Plain bars show the value of each metric at the finest taxonomic level, striped**
**bars show the value after regrouping objects into broader ecological groups. All values, including F1-scores, are reported in Table S8.**

Moving from native features to MobileNet deep features before the RF classifier significantly increased all classification metrics (Fig. 4). On the contrary, performance stayed the same when the MLP600 classifier was replaced by a RF after the

same MobileNet feature extractor. This suggests that the classifier itself is of relatively little importance; rather, it is the quality of the features that determines performance. Since features are optimized during CNN training, their quality aligns with the patterns the algorithm learns to improve classification accuracy.

Finally, compressing features with a classification-agnostic dimension reduction method (PCA here) had very little effect on classification performance (Fig. 4). This supports the idea, stated in the previous section, that the information required to classify the relatively small, gray-scale plankton images captured by the instruments considered here can be efficiently summarized in only a few numbers (50 here). This opens operational possibilities since the feature extractor, the feature compressor and the classifier can be separated.

### 3.6 Performance on coarser groups

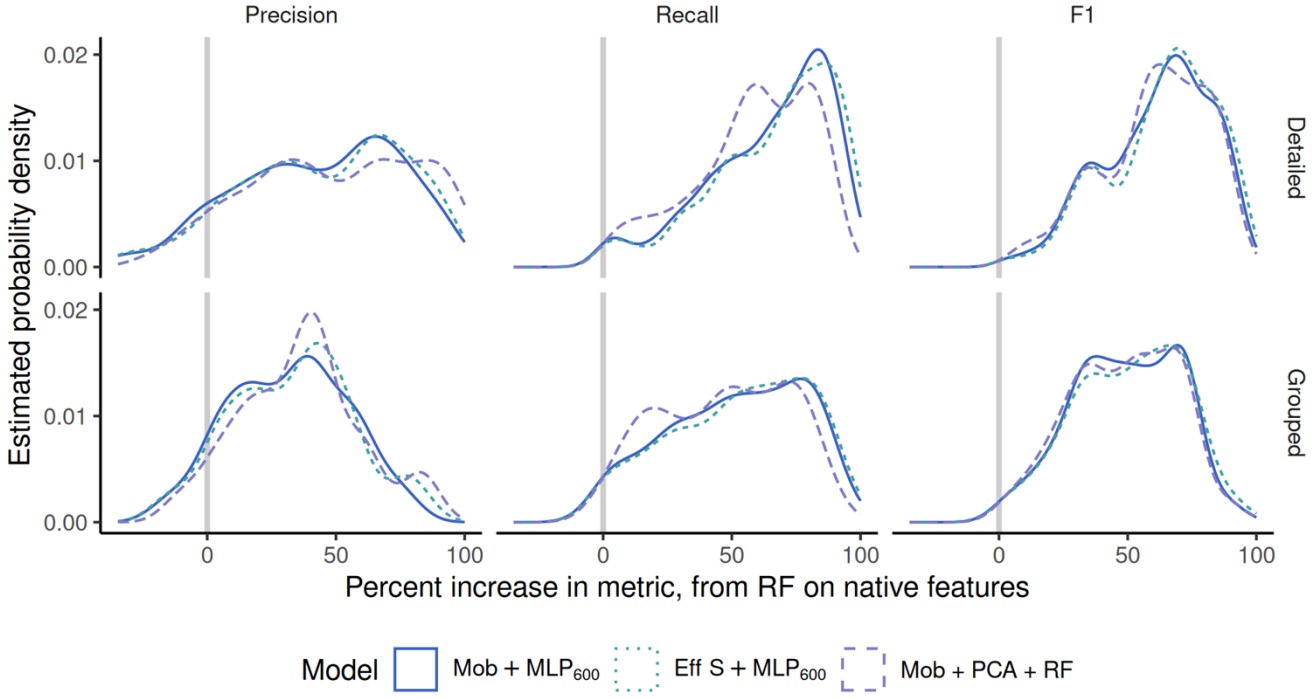

Figure 5: Density distribution (i.e. continuous histogram) of the difference in performance metrics per class when going from RF on native features to different deep models (colors), on the ZooScan datasets, at two taxonomic levels (rows).

Regrouping classes into broader ecological groups improved all performance metrics (accuracy, plankton precision and plankton recall) across all datasets and approaches (Fig. 2, 3, and 4), as it made the classification task easier, in line with previous results (Kraft et al., 2022). However, it is important to note that our method − regrouping classes after training on detailed classes − differs from retraining a model on grouped classes alone. In the latter approach, regrouping would increase

the number of examples within each group, likely enhancing performance. Yet, this could also introduce more diversity within each class, sometimes referred to as "within-class subconcepts" (He and Garcia 2009), which might reduce accuracy in certain, morphologically diverse, groups (e.g. both Appendicularia bodies and houses being labeled as Appendicularia). This decrease in performance is especially evident in miscellaneous classes containing objects that could not be assigned to other categories (Tables 4, S2 - S7). The performance increase between detailed and coarse classes was larger for classical approaches, particularly on the ZooCam and ZooScan datasets (Fig. 2). This highlights the fact that classical approaches often confused fine-scale taxa, comprised within larger groups. A good example is Copepoda, which has 22 subclasses in the ZooCam dataset and 20 in the ZooScan dataset. The classification of some of these ~20 classes was often poor with classical models while the classification of Copepoda, as a whole, was rather good. Since Copepoda represented a large percentage of the images in each dataset, 38% and 34% respectively, classifications metrics significantly improved when they were grouped.

The other side of the same coin is that performance improvements when going from a RF on native features to different deep models were larger when the taxonomic level was more detailed. In Fig. 5, most classes show better performance with the deep models (to the right of zero), and the increase is more pronounced with detailed classes (top) than on regrouped ones (bottom), for precision in particular. In other words, deep models beat classical ones on almost all classes (most differences in per-class metrics were above zero) but, on datasets with more and smaller classes, CNN beat classical approaches more often and by a wider margin than on coarser datasets. This further supports that CNN are better than classical approaches specifically at classifying rare classes.

## 4 Discussion

### 4.1 Costs and benefits of using CNN

In terms of accuracy alone, CNN did not appear to offer a significant performance improvement over the classical approach of handcrafted feature extraction followed by a RF classifier. However, the high scores of a purely random classifier on this metric show how flawed it can be on unbalanced datasets. Instead, balanced accuracy (Kelleher et al. 2020) and metrics on plankton classes only both showed that CNN performed better in classifying objects, especially in low abundance classes (and when class weights were used). This was further confirmed by the fact that the difference between CNN and the classical approach was lower when classification was performed at a coarser taxonomic level. This makes the use of pretrained CNN particularly relevant for plankton images classification, which are particularly diverse, contain many small classes and in which the dominant classes are often composed of various detritus and artifacts.

Giving more weight to poorly represented classes resulted in better performance, especially for RF. One plausible explanation would be that weighted RF (Chen et al. 2004) actually make use of class weights twice: weights are used to

compute the criterion to generate the splits (entropy in our case) when building the tree; weights are also used when voting for the majority class in terminal nodes. On the other hand, class weights are only used to compute a weighted loss in CNN (Cui et al. 2019).

While CNN took longer to train than RF in terms of overall training duration, the comparison is not straightforward. First, training a RF model requires extracting features from the images beforehand. This feature extraction is coded, not trained, so this part cannot be directly compared. Additionally, it can be challenging to know when feature extraction is truly complete, as the optimal set of features often depends on the specific dataset and task. But even in terms of pure evaluation (i.e. extracting features and predicting the class of new images), the computation of some handcrafted features can take a non-negligible amount of time and a CNN may prove faster, notably thanks to the use of GPUs by the underlying software libraries (Chellapilla et al. 2006). Additionally, the training time of CNN depends heavily on the number of parameters. For instance, our lightweight model (MobileNet V2) trained in under 100 hours, which is fast compared to larger models (Zebin et al. 2019). Since lightweight CNN models demonstrated performance comparable to larger ones for plankton classification tasks (e.g. Kraft et al., 2022), they present an appealing choice: their computational demands are often modest and compatible with most recent computers. Finally, a metric that may be more relevant than computational time for many applications is the total time investment of the scientific team, including model setup, training, and output validation. In this respect, we argue that CNN are actually simpler to adopt. Modern deep learning libraries such as Tensorflow (Abadi et al. 2016) or Pytorch (Paszke et al. 2019) are free and open-source, and the abundance of tutorials and pre-trained models means that users need little image processing or coding expertise to get started, whereas extracting relevant handcrafted features typically requires domain-specific knowledge. Although training a CNN may involve some technical steps (e.g. configuring a data loader), the deployment stage is extremely lightweight, often only a few lines of code to load the saved model and run inference. Consequently, the resulting model packages the whole pipeline (from image pre-processing to classification) and can be deployed on various devices. And as GPU resources become increasingly available for the scientific community, these powerful tools become more accessible (Malde et al. 2020).

Finally, our results highlight the efficacy of both CNN and classical methods for accurate prediction of well-represented plankton classes. However, rare classes still require manual validation by a taxonomist. Importantly, improved prediction quality achieved by CNN compared to classical approaches is likely to save time by reducing the need for prediction corrections, as reported by Irisson et al. (2022).

## 4.2 Importance of the quality and number of features

Models using a CNN feature extractor, which generated features much more numerous than the handcrafted ones (>1000 vs. ~50), performed better as expected from the literature (Orenstein and Beijbom 2017). Increasing the size of the feature extractor, hence yielding potentially richer features (keeping their number in the same order of magnitude: 1792 for the

MobileNet V2 vs. 1280 for the EfficientNet V2) did not lead to a significant improvement in classification performance; but it did lengthen the training time. Reducing the number of features from a CNN to an amount similar to the number of handcrafted features (50), using PCA or compression within a small fully connected layer, did not significantly affect classification performance either. These results show that the richness and diversity of features is important, but only to a certain extent with plankton images. Although features from CNN cannot be individually interpreted, texture features were shown to be important for image classification by CNN (Baker et al. 2018). Moreover, visualization techniques have been developed to provide insights into the convolutional layers of CNN, revealing that convolutional layers detect patterns like edges and textures (Zeiler and Fergus 2014). By contrast, most handcrafted feature sets were poor in texture-related features, which may explain their lower performance.

The fact that the number of features can be greatly reduced (from 1792 to 50, a 36-fold reduction, in our case; from 216 to 25, an 8-fold reduction, in Guo et al. 2021b) suggests there is only a limited amount of relevant information in plankton images for CNN to extract. These images are typically small (~100×100 pixels for the average ZooScan image) and often grayscale, which restricts the amount of useful information available to any classifier. Consequently, increasing network depth or size does not yield appreciable performance gains, because the intrinsic information in the images is already fully exploited by a small CNN.

Therefore, improvements in classification accuracy are more likely to come from richer inputs than from larger network architectures. One way to achieve this is by increasing the quantity of annotated plankton images; pooling data from multiple instruments and sampling conditions has been shown to improve CNN accuracy (Ellen and Ohman, 2024) and this is the first step towards building a so-called foundation model for plankton images. A second, independent route is to enhance the informational content of each image. For example, color cameras such as those used in the planktoscope (Pollina et al. 2022) or the Scripps Plankton Camera (Orenstein et al. 2020b), should capture more information by using multiple channels. Beyond color, additional fluorescence channels can be obtained using environmental high content fluorescence microscopy, enriching the information content of images (Colin et al. 2017); but this method can only be applied ex situ. Expanding the amount of training data and capturing richer image information should both yield gains in classification performance, albeit at the cost of greater storage and processing requirements. Our findings also open an opportunity to simplify plankton image classification models, by performing a wise feature selection through recursive feature elimination for example (a backward selection of less informative features until only informative features remain; Guyon et al. 2002; Guo et al. 2021b). Dimension reduction techniques, such as PCA (Legendre and Legendre 2012), can also be used to remove both correlations and noise in the features. The combination of deep feature extraction, dimension reduction, and a robust classifier, such as RandomForest, is lightweight and quick to train, yet yields high quality results (Fig. 4). Because of these advantages, this approach has been implemented in the EcoTaxa web application (Picheral et al. 2017), allowing users to apply such methods to their own plankton image datasets.

The similar performance between a full CNN and a deep feature extractor combined with a RF classifier (Fig. 4) suggests that the nature of the features is much more important than the nature of the classifier. These results are consistent with those comparing different classifiers on handcrafted features, where no significant differences could be highlighted (Grosjean et al. 2004; Blaschko et al. 2005; Gorsky et al. 2010; Ellen et al. 2015). Still, in highly unbalanced datasets (IFCB, ISIIS and UVP6), the plankton precision was slightly higher with the RF than with the $MLP_{600}$, reflecting a lower contamination of plankton classes by dominant detritus. Its stronger sensitivity to class weights is another possible explanation in our case.

### 4.3 Alternative approaches for plankton image classification

A potential drawback of CNN is that they may not account for the real size of objects, since all images are rescaled to the same dimensions before input. One solution to capture size would be not to scale down images larger than the input dimension but to pad the smaller ones with the background color. However, very small objects may be reduced to just 1 pixel after a few pooling layers and all information in the original image could be lost. Another common solution would be to concatenate size information from handcrafted features (e.g. area, Feret diameter) or simply the image diagonal size to one of the fully connected layers to create a model that accounts for both image aspect and object size. Still, despite the a priori relevance of size to recognize plankton taxa, such approaches do not necessarily provide a large improvement in classification performance: Kerr et al. (2020) report a small improvement when geometric features are concatenated, while Kyathanahally et al. (2021) report a negligible gain. Ellen et al. (2019) evaluated the effect of concatenating different types of "metadata" (geometric, geotemporal and hydrographic) to fully connected layers: geometric features alone did not improve model performance, whereas geotemporal and hydrographic metadata each yielded a noticeable boost, and adding geometric metadata on top of those provided an additional improvement. One possible explanation is that deep features already capture the essential information needed for classification, making additional geometric features redundant. However, adding geotemporal and hydrographic features (individually or combined) enhanced prediction performance, which is unsurprising given the patchy nature of plankton organisms. Plankton taxa tend to exhibit positive correlations within groups (Greer et al. 2016; Robinson et al. 2021), and are often associated with specific environmental parameters—a relationship that machine learning algorithms can leverage (e.g., relating plankton biomass to environmental conditions, as shown in Drago et al. 2022). However, one should keep in mind that incorporating metadata features during training may hinder subsequent analyses linking these organisms to their environment, since the classifier learned a correlation between the abundance of some organisms and some environmental conditions from the training set, and will therefore induce it in its predictions.

As highlighted above, plankton datasets are often highly unbalanced, with few objects in plankton classes while the largest classes often consist of non-living objects such as marine snow. There are both "algorithm-level" and "data-level" methods for dealing with class imbalance (Krawczyk 2016), which can be used separately or simultaneously. Algorithm-level

methods include the use of class weights to give more importance to poorly represented classes in the loss computation (Cui et al. 2019); like we did here. Another algorithm-level method is to use a different loss function, such as sigmoid focal cross entropy (Lin et al. 2018), which penalizes hard examples (small classes) more than easier ones (large classes). Data-level methods include oversampling small classes and undersampling large classes, thereby rebalancing the distribution of classes in the training set (Krawczyk 2016). While this practice often improves performance on a test set to which the same modifications are applied, it can lead to poor performance when evaluating the model on a real, therefore unbalanced, dataset, because the model has learned an unrepresentative class distribution from the training set. This problem is known as "dataset shift" (Moreno-Torres et al. 2012). Typically, using a model trained on an idealized training set to classify objects from a new, real dataset leads to poor prediction quality (González et al. 2017). Similarly, a model trained for specific conditions (such as location, depth, or time) will likely fail to generalize to images acquired under different circumstances. To mitigate this, a potential solution would be to assemble a training set from samples that match the context of the future deployment (similar climate and season), hoping that similar context will give rise to similar class distributions. Alternatively, and more generically, the training set can be made as exhaustive as possible by spanning a wide range of spatial and temporal conditions; its global class distribution would minimize the average differences with the class distribution of new samples. Consequently, the impact of the dataset shift depends directly on how representative the training data are of the spatial and temporal regimes of interest. All types of classification models, including cutting-edge architectures like vision transformers, are susceptible to dataset shift (Zhang et al. 2022). Today, there is no obvious solution to deal with dataset shift in classification tasks and other approaches, such as quantification, should be considered (González et al. 2019; Orenstein et al. 2020a).

*Weighting improves the recall of rare classes but reduces their precision, reflecting the classic precision–recall trade-off. When downstream analysis involves manual verification, higher recall is advantageous because a few false positives in rare classes can easily be corrected while missed detections would likely be lost among the most numerous classes and not easily recovered. Conversely, in high-throughput monitoring through imaging, where human review of all samples is infeasible, emphasizing precision reduces spurious detections at the cost of under-estimating true abundances. In such settings, post-hoc confidence thresholding (e.g. Faillettaz et al., 2016; Luo et al., 2018) offers a pragmatic compromise, albeit an imperfect one. In all situations, using various intensities of class weighting is a flexible solution to adapt the classifier to the study's objective*

The rarity of some plankton classes means that some classes will inevitably be absent from the training set. Because a conventional classifier is trained on a fixed label list, every object is forced into one of these known classes, causing novel or poorly characterized organisms to be misclassified. In these situations, approaches such as unsupervised, self-supervised or semi-supervised learning (e.g. autoencoders) or specific open-set classifiers can be employed (Bendale and Boult, 2016;

Ciranni et al., 2025; Masoudi et al., 2024). These methods can leverage the rich feature embeddings produced by a CNN while detecting objects that do not belong to any of the known training classes (Malde and Kim 2019; Schröder et al. 2020).

## 5 Conclusion and perspectives

In summary, a small CNN achieved strong performance at plankton image classification across six realistic plankton image datasets, while being easy to apply. It unsurprisingly outperformed the classical approach of extracting a small number of handcrafted features and using a RF classifier, particularly for rare classes. Applying per-class weighting improved the detection of underrepresented classes. Surprisingly, using a large CNN did not lead to better classification performance than a much smaller one and deep features could be quite heavily compressed without loss of performance. This is likely related to the fact that plankton images, which are typically small and grayscale, provide relatively little information content for CNN. Richer images (e.g. higher resolution, colour or multispectral data) produced by next-generation imaging systems would provide additional discriminative information that bigger models could leverage. Finally, the nature of the features dominated the outcome: deep features drove the performance gains, while the choice of classifier had little impact. Overall, these findings suggest that larger and more diverse training sets and/or advances in imaging hardware, rather than ever larger models, will be key to further improving plankton classification. Furthermore, metrics that emphasize the classes of interest − often the minority classes in plankton datasets − should be prioritized.

The results presented here are in line with the shift towards the use of deep learning models for plankton classification tasks (Rubbens et al. 2023), which was made possible by advances in computational performance through easier access to dedicated hardware, the release of sufficiently large datasets, and the development of turnkey deep learning libraries such as Tensorflow (Abadi et al. 2016) or Pytorch (Paszke et al. 2019). Datasets in this study are made publicly available to facilitate future benchmarking of new classification methods.

## Data availability

The datasets used in this study are presented in Table 5.

**Table 5: References of datasets used in this study.**

| Dataset name | Reference | DOI | URL |
|---|---|---|---|
| IFCB | Sosik et al. 2015 | 10.1575/1912/7341 | https://doi.org/10.1575/1912/7341 |
| ISIIS | Panaïotis et al. 2024 | 10.17882/101950 | https://doi.org/10.17882/101950 |
| FlowCAM | Jalabert et al. 2024 | 10.17882/101961 | https://doi.org/10.17882/101961 |
| UVP6 | Picheral et al. 2024 | 10.17882/101948 | https://doi.org/10.17882/101948 |
| ZooCAM | Romagnan et al. 2024 | 10.17882/101928 | https://doi.org/10.17882/101928 |

## Code availability

All the code supporting this study is available at https://doi.org/10.5281/zenodo.17937437 (Panaïotis and Amblard 2025).

## Author contribution

JOI and TP conceived the study; GBC and GDA developed a first CNN classifier; TP and EA implemented the RF classifier
640 and the final CNN classifier from the initial work of GBC and GDA, with guidance from BW; EA performed the
experiments under the supervision of TP and JOI; TP wrote the original draft; all authors reviewed and approved the final
manuscript.

## Competing interests

Emma Amblard was employed by Fotonower. Guillaume Boniface-Chang was employed by Google Research, London.
645 Gabriel Dulac-Arnold was employed by Google Research, Paris. Ben Woodward was employed by CVision AI.

## Acknowledgement

We would like to acknowledge scientists, crew members and technicians who contributed to data collection and the
taxonomist experts who sorted the images to build the datasets. Special thanks go to Eric Orenstein for providing scripts to
extract handcrafted features from IFCB images and for his valuable feedback on the manuscript.

650 ## Financial support

This work was carried out within the projects "World Wide Web of Plankton Image Curation", funded by the Belmont
Forum through the Agence Nationale de la Recherche ANR-18-BELM-0003-01 and the National Science Foundation (NSF)
ICER1927710, and LOVNOWER funded by the program "France relance" from December 21st 2020. TP's doctoral
fellowship was granted by the French Ministry of Higher Education, Research and Innovation (3500/2019). This work was
655 granted access to the HPC resources of IDRIS under the allocation AD011013532 made by GENCI. TP was supported by
projects CALIPSO funded by Schmidt Sciences and BIOcean5D funded by Horizon Europe (101059915). Views and
opinions expressed are those of the author(s) only and do not necessarily reflect those of the European Union. Neither the
European Union nor the granting authority can be held responsible for them.

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
