# Peer review of "Benchmark of plankton images classification: emphasizing features extraction over classifier complexity"

_Earth System Science Data, 2025_

## Author Comment (AC1)

**Responses to reviewers**

**Benchmark of plankton images classification: emphasizing features extraction over classifier complexity**

We thank the editor and the reviewers for their constructive feedback. Below, we provide responses (in blue text) to each comment in support of our revised submission. Please note that the line numbers in our responses refer to those in the revised manuscript with tracked changes. A version of the manuscript without tracked changes has been uploaded separately.

**Editor**

1) There seems to be a typo on your pdf regarding your co-author: it must be "Boniface-Chang" instead of "Boniface-Change". 2) Your table 4 contains coloured cells. Please note that this will not be possible in the final revised version of the paper due to HTML conversion of the paper. When revising the final version, you can use footnotes or italic/bold font. For now, the process will continue, but please note that the final version cannot be published by using coloured tables. 3) Please make sure to also include the dataset DOI of the respective references in the reference list.

Thank you for your helpful comments. We have corrected the co-author's name to Boniface-Chang. All colored cells have been removed from Table 4 in the main manuscript; the original colored version is now provided as Supplementary Table S7. In the Data Availability section we have replaced the paragraph with a table that lists datasets used in this study together with reference, DOI and URL.

**Reviewer 1**

There is a huge number of studies conducted on the topic of plankton classification. The authors of the manuscript are trying to provide a baseline comparison dataset that different methods could be compared against, and a dataset that could be used in order to enable better comparison of results between the numerous studies. The aims of the study have good grounds, giving justification for the study. The authors present an interesting case with a well-designed manuscript. There are two minor topics of improvements: the authors emphasize and acknowledge only large datasets, in my opinion, smaller efforts in trying to provide public datasets should also be given some acknowledgement in the introduction, and another minor issue is in the way the authors present alternative methods, with for

example no mentioning on open set classification methods and e.g., autoencoders, that have been demonstrated as rather promising methods for different tasks.

We thank you for the thorough and encouraging review. We are pleased that you find the motivation, design and overall presentation of the manuscript strong. We agree with the two limitations you pointed out, and have revised the manuscript accordingly. Further below, we also address all the specific points you raised.

In the introduction, we have modified the paragraph at line 166 to acknowledge the role of smaller public plankton datasets:

"This suggests that classifiers improved, although this is unquantifiable for all the reasons above. Earlier plankton image datasets were modest in size, typically containing a dozen or a few dozen of classes (Benfield et al., 2007), but were crucial for establishing the first classification methods. Building on that foundation, three major plankton image datasets have been published and used in several studies (Table 1) [...]"

Additionally, we have revised the last paragraph of the "Alternative approaches for plankton image classification" Discussion subsection to mention promising approaches such as open-ended classification and autoencoders. The text at line 714 now reads as follows:

"The rarity of some plankton classes means that some classes will inevitably be absent from the training set. Because a conventional classifier is trained on a fixed label list, every object is forced into one of these known classes, causing novel or poorly characterized organisms to be misclassified. In these situations, approaches such as unsupervised or semi-supervised representation learning (e.g. autoencoders) or specific open-set classifiers can be employed (Bendale and Boult, 2016; Ciranni et al., 2025; Masoudi et al., 2024). These methods can leverage the rich feature embeddings produced by a CNN while detecting objects that do not belong to any of the known training classes (Malde and Kim 2019; Schröder et al. 2020)."

Specific comments with numbers referring to lines:

**53-55: Could add some references to demonstrate the imbalance**

We thank the reviewer for raising this point. Such references have been inserted at line 60: "To begin with, planktonic communities (Ser-Giacomi et al., 2018), and therefore the resulting image datasets (Eftekhari et al., 2025; Schröder et al., 2019), exhibit significant class imbalance."

57-58: True, but also, if a class has very distinguishable morphology, the required number of training images for the class to perform well will be much less. See e.g. Kraft et al. 2022 https://doi.org/10.3389/fmars.2022.867695

Thank you for raising this. The text has been updated accordingly at line 65:

"As a consequence, rare planktonic classes are usually harder to predict for automated algorithms (Lee et al. 2016; Van Horn and Perona 2017; Schröder et al. 2019), although classes with highly distinctive morphologies could still be correctly classified even with few training images (Kraft et al. 2022)"

83-84: There is a mention of plankton traits, but the topic of traits has not been touched on previously in the introduction and would require a description earlier.

Thank you for raising this issue. A brief definition of traits has been inserted at line 85:

"These manually extracted features – intended to capture plankton traits (observable characteristics, primarily morphological) – aim to summarize the image content in numerical form, providing a concise representation that facilitates the classification process."

120-122: Two recent review articles are covered, but you are missing a third, even more recent review on plankton classification methods, Eerola et al. 2024 https://doi.org/10.1007/s10462-024-10745-y , in particular fig 3.

Thank you for pointing us to this recent review. We have incorporated it at line 146:

"Currently, research on the classification of plankton images, or images of any other type of marine organisms, is dominated by CNN (Irisson et al. 2022; Rubbens et al. 2023, Eerola et al., 2024)."

135-136: This is especially true for studies that concentrate on machine learning methods. If looking at the studies that concentrate more on the implications of the results in ecological, taxonomical, or operational contexts, the class-specific metrics are more often published. How did you come up with the list in Table S1? The 10 most cited are based on the Irisson et al. 2022, right? Is the rest cited in this paper the total of citations in your manuscript (sorry, I was too lazy to count them)? If so, are you sure they are representative of the entire literature on the topic, as you are not covering all recent publications with plankton recognition using CNNs? I would not draw this type of conclusion unless you have tried to ensure you cover all recent publications. A number of studies has been published since Irisson et al. 2022.

Thank you for this comment. As indicated at line 163, Table S1 does list the references that are cited in the manuscript; the first ten rows correspond to the ten most-cited papers identified by Irisson et al. (2022). In the revised version we have added the additional references that were introduced during the revision and performed a literature check to verify that recent plankton-image classification studies were included. The updated Table S1 now includes those new citations.

297-299: Why is this? It goes against the gut feeling, so in addition to references, could you also mention why?

We thank the reviewer for raising this point. The text has been updated at line 380 to explain this issue:

"Although both types of methods were shown to improve classification performance in some situations (e.g. for a binary classification task, McCarthy et al. 2005), resampling forces the model to learn on an artificial, balanced class distribution; when the real-world data have a different (often skewed) distribution, the learned decision thresholds become mis-calibrated and performance degrades (Moreno-Torres et al. 2012; González et al. 2017)."

335: change the term pure into something else e.g., how false positive-free the predicted plankton classes are

Thank you for this suggestion, the text has been updated at line 421 and now reads:

"Averaged plankton recall gives a direct indication of the proportion of planktonic organisms that were correctly predicted, while averaged plankton precision reflects how free the predicted plankton classes are from false positives."

Table 4: It would be good to add the class-specific n for each class, as it most often will/can, but not exclusively, affect/explain the class-specific performance. And also, to the tables with other datasets. What means the Plankton? Do the colors mean something? Please add more information to the table caption. Yes, after scrolling down, there is also non plankton. Could the word classes be added after those, i.e., Plankton classes and Non plankton classes? I didn't find the corresponding tables for the other datasets, where are they?

We thank the reviewer for the comment. Table 4 has been updated to include the number of objects (N test) for each class in the test set. The cell colours were originally intended to reflect the values inside the cells; however, the editor informed us that coloured cells cannot be retained in the final version, so the shading has been removed. We also added the word "classes" after "Plankton" and after "Non-plankton" to make the distinction clear. Tables for other datasets are provided in the supplementary material (Tables S2 to S7) and have been updated accordingly.

Header 3.3: Please change this from revealing the results into something like Model performance on small classes

We thank the reviewer for the suggestion. We chose this heading deliberately because it gives readers an immediate sense of the main observation discussed in the subsection; stating the outcome up front provides a concise guide to the part of the paper that addresses the performance gap on under-represented classes. Consequently, we prefer to retain the original wording.

Figure 2: Why did you choose to show accuracy and not F1 score in the first panel (the same comment goes also for the subsequent figures)? What is the Random classifier? It was mentioned in a paragraph starting from line 350, but it would require a better explanation. We thank the reviewer for this comment. Accuracy is plotted in the first panel to illustrate how a naïve baseline can be misleading on highly imbalanced plankton datasets. By displaying the random-classifier accuracy alongside the model accuracies, the figure makes the gap between a trivial solution and our trained models immediately apparent. The harmonic mean of precision and recall (F1) is already reported for each class in Table 4 and in the supplementary tables (S2–S7); therefore adding it again in the first panel would be redundant.

Regarding the random classifier, it is defined in lines 408-415. It assigns class labels according to the empirical class frequencies of each dataset but randomly, providing a realistic lower bound for performance on imbalanced data. For example, on the UVP6 dataset the random baseline attains 63 % accuracy (Figure 2). Reporting this figure prevents a superficial interpretation of a modest (e.g. 70 %) accuracy as "good" when, in fact, it is only marginally better than chance. Furthermore, the random classifier produces a non-trivial confusion matrix (as opposed to a null classifier that would assign all objects to the dominant class), allowing the computation of class-specific metrics. The random baseline thus offers a transparent reference point for evaluating all subsequent models. The text has been updated to better emphasize this aspect at line 408:

"For instance, in an 11-class dataset with one dominant class comprising 90% of the data (and each of the other classes making up only 1%), a classifier that always predicts the dominant class would achieve 90% accuracy but would provide no insight into the ten minority classes. A random classifier that draws labels according to the empirical class distribution would yield a lower-bound 81% accuracy  $(0.9^2 + 10 \times 0.01^2)$ . This baseline reflects the underlying distribution while still producing a full confusion matrix that can be used to compute metrics such as precision and recall."

375-385: Wouldn't it be important to find a harmonic mean between precision and recall rather than emphasize the importance of precision and detection of rare classes over recall? Thank you for this comment. Our primary goal in this section is to illustrate how class-weighting influences the two components of performance: precision (contamination of the predicted plankton classes) and recall (ability to detect under-represented plankton). Because the datasets are highly imbalanced, the trade-off between these aspects is central to the methodological discussion: weighting decreases false negatives for scarce plankton classes (raising recall) at the cost of more false positives (lowering precision). Presenting precision and recall separately makes this trade-off explicit and lets readers see how the weighting scheme shifts the error distribution.

The relative importance of precision versus recall depends on the downstream ecological question. For studies that need reliable detection of rare taxa, higher recall is crucial even if precision drops; for abundance-focused work, higher precision may be preferred. Since the optimal balance is study-specific, we chose to report the two metrics individually rather than collapse them into a single F1 score.

Header 3.4: I would rephrase this as well rather to be i.e., Model performance of a small CNN in plankton image classification

Thank you for this suggestion. As for header 3.3, we wish to retain headers which guide the readers towards the main findings.

Header: 3.5: Importance of features and classifier

Thank you for this suggestion. As for headers 3.3 and 3.4, we prefer to retain original headers.

420-421: You mean recall and precision? You did not show F1 in the figures you are referring to.

We thank the reviewer for the comment. The sentence at lines 420-421 refers specifically to the metrics shown in Figures 2, 3 and 4 (accuracy, balanced accuracy, plankton precision and plankton recall). No F1 scores are presented in those figures, nor are they mentioned in these lines. To avoid any ambiguity we have revised the wording at line 529:

"Regrouping classes into broader ecological groups improved all performance metrics (accuracy, plankton precision and plankton recall) across all datasets and approaches (Fig. 2, 3, and 4), as it made the classification task easier [...]."

421-423: This is in line with the results from Kraft et al. 2022 where there was almost no confusion between different taxonomical groups.

Thank you for raising this. This reference has been included at line 529:

"Regrouping classes into broader ecological groups improved all performance metrics (accuracy, plankton precision and plankton recall) across all datasets and approaches (Fig. 2, 3, and 4), as it made the classification task easier, in line with previous results (Kraft et al., 2022)"

467-469: A lightweight CNN has proven to reach very good performance in classifying plankton also previously, e.g., Kraft et al. 2022.

Thank you for pointing us to this relevant work. The reference has been inserted at line 578: "Since lightweight CNN models demonstrated performance comparable to larger ones for plankton classification tasks (e.g. Kraft et al. 2022), they present an appealing choice: their computational demands are often modest and compatible with most recent computers."

470-475: Yes, I do agree with this partly, however, for example, the MATLAB codes available for IFCB data processing and classification purposes are still easier to adopt by new groups, as they don't actually require much knowledge of any programming. That is why so many groups with IFCB still actively use the MATLAB-based RF implementation <a href="https://github.com/hsosik/ifcb-analysis">https://github.com/hsosik/ifcb-analysis</a>

Thank you for this comment. We agree that this specific IFCB MATLAB-based pipeline is user-friendly, however a practical advantage of modern deep-learning approaches is that they rely on free, open-source Python libraries (TensorFlow, PyTorch), whereas MATLAB requires a commercial license. These tools also come with tutorials and pre-trained models, allowing researchers with little image-processing experience to train and deploy CNN with only a few lines of code. The paragraph at line 581 has been updated as follows:

"In this respect, we argue that CNN are actually simpler to adopt. Modern deep-learning libraries such as TensorFlow (Abadi et al., 2016) or PyTorch (Paszke et al., 2019) are free and open-source, and the abundance of tutorials and pre-trained models means that users need little image-processing or coding expertise to get started, whereas extracting handcrafted features typically requires domain-specific knowledge. Although training a CNN may involve some technical steps (e.g. configuring a data loader), the deployment stage is extremely lightweight, often only a few lines of code to load the saved model and run inference. Consequently, the resulting model packages the whole pipeline (from image pre-processing to classification) and can be deployed on various devices."

510-511: The comment comes a bit out of the blue and without context. If you want to add this information, I suggest rephrasing and tying it better to the content.

We thank the reviewer for this suggestion. The text has been updated at line 635:

"Because of these advantages, this approach has been implemented in the EcoTaxa web application (Picheral et al. 2017), allowing users to apply such methods to their own plankton image datasets."

545-547: Do you have statistics/ figure to support this? If it is said like this, the results should be added as supplementary, otherwise, this phrase should be removed.

Thank you for raising this issue. Experiments with focal cross entropy were done very preliminary and abandoned given the absence of performance improvements. We thus do not have publication ready results to illustrate this. We have removed the sentence accordingly at line 690.

548-550: The concept dataset shift is indeed a problem. However, the nature of plankton makes it very hard to have a representative distribution when classifying real datasets. The training data is ideally constructed based on data from multiple occasions, seasons, and covering several years. Still, when classifying data, the samples to be classified are from a specific time point, i.e., a spring sample, which will not have the same data distribution as the training data, as the class composition and the share of very heterogeneous images vary. So, an interesting question is, how much does the class distribution actually matter in the case of plankton?

Thank you for raising this point. Plankton communities are indeed intrinsically dynamic and heterogeneous. Consequently, the degree to which class-distribution mismatch affects performance is highly dataset-specific. We have updated the text at line 702:

"Similarly, a model trained for specific conditions (such as location, depth, or time) will likely fail to generalize to images acquired under different circumstances. To mitigate this, a potential solution would be to assemble a training set from samples that match the context of the future deployment (similar climate and season), hoping that similar context will give rise to similar class distributions. Alternatively, and more generically, the training set can be made as exhaustive as possible by spanning a wide range of spatial and temporal conditions; its global class distribution would minimise the average differences with the class distribution of new samples."

555-561: Open set classification methods are also a very promising approach to classify plankton, as plankton data includes a lot of difficult-to-classify images that normally end up in very heterogeneous classes with little common good features to describe those classes, and which often have very poor performance. Still, often those classes, at least in the case of phytoplankton instruments triggered by chlorophyll a, contain phytoplankton, i.e., are of interest, but which are difficult or impossible to identify taxonomically well (e.g., a small flagellate).

We thank the reviewer for the suggestion. We agree that open-set classification is a promising direction for plankton imagery. Accordingly, we have expanded the paragraph to acknowledge this approach explicitly at line 714:

"The rarity of some plankton classes means that some classes will inevitably be absent from the training set. Because a conventional classifier is trained on a fixed label list, every object is forced into one of these known classes, causing novel or poorly characterized organisms to be misclassified. In these situations, approaches such as unsupervised or semi-supervised representation learning (e.g. autoencoders) or specific open-set classifiers can be employed (Bendale and Boult, 2016; Ciranni et al., 2025; Masoudi et al., 2024). These methods can leverage the rich feature embeddings produced by a CNN while detecting objects that do not belong to any of the known training classes (Malde and Kim 2019; Schröder et al. 2020)."

570-572: I don't see this as a new and interesting thing, but rather an already well-known fact. I didn't see the point of showing accuracy in figures as well, as the fact that it is a poor metric is already known.

We thank the reviewer for the comment. Global accuracy is indeed an inadequate metric for highly imbalanced datasets, yet it continues to be reported in many plankton-ecology studies, largely because it is the standard measure used in image-classification challenges. As shown in Table S1, only about half of the cited studies report performance metrics beyond global and/or averaged metrics (most of the time only accuracy, sometimes recall, precision or F1), underscoring the need to remind readers of its limitations and to promote measures that give proper weight to minority classes. For example, Eerola et al. (2024) report accuracy values on a variety of publicly available datasets, showing that accuracy remains a widely used metric despite its well-known limitations.

**Reviewer 2**

While I do not think the paper is original, because the concept is simple, it is well overdue and very clearly presented. The data quality is Excellent in terms of diversity of instruments represented, but only good in terms of size of datasets (although that is limited public data sets). Most of my comments are based on opinions and setup, not the actual methodology of the manuscript itself.

Thank you very much for your thorough and constructive review. We appreciate the recognition that the manuscript is clearly presented and that the dataset spans a wide variety of instruments, which was our main goal from the start. Below we address each of the points you raised.

Line 26 - I am not following the logic of how the finding of small CNNs performing sufficiently well at classification of small grayscale plankton images enables imaging systems to provide larger images. The quality of the images is dependent upon the physics/optics and electrical engineering, not the ML algorithm.

We thank the reviewer for this comment. Our intention was to highlight that the limited benefit of larger CNN stem from the intrinsic information content of the current plankton images, not from a deficiency of the deep learning algorithms. Because the images are small ( $\approx 100 \times 100 \text{ px}$ ) and often grayscale, a compact CNN already extracts essentially all usable signal, and adding depth or increasing the number of parameters yields no measurable gain. From this observation we infer that substantial performance improvements are unlikely to come from more sophisticated ML models alone. Instead, enhancing the raw data (e.g. higher-resolution, color images) would increase the amount of discriminative information available to any classifier, including CNN. In other words, the bottleneck is the physical quality of the images, and upgrading the imaging system is the most promising route to further gains.

**We have updated the text at line 614:**

"The fact that the number of features can be greatly reduced (from 1 792 to 50, a 36-fold reduction, in our case; from 216 to 25, an 8-fold reduction, in Guo et al. 2021) suggests there is only a limited amount of relevant information in plankton images for CNN to extract. These images are typically small ( $\sim 100 \times 100$  pixels for the average ZooScan image) and often grayscale, which restricts the amount of useful information available to any classifier. Consequently, increasing network depth or size does not yield appreciable performance gains, because the intrinsic information in the images is already fully exploited by a small CNN."

**And at line 724:**

Surprisingly, using a large CNN did not lead to better classification performance than a much smaller one and its deep features could be quite heavily compressed without loss of performance. This is likely related to the fact that plankton images, which are typically small and grayscale, provide relatively little informative content for CNN. Richer images (e.g. higher resolution, color or multispectral data) produced by next-generation imaging systems would provide additional discriminative information that bigger models could leverage.

Line 50 - I disagree that the software pipelines have not progressed as fast as the hardware. I think the case made in Malde et al. is that the analysis is not keeping up with either the software or hardware. For example, if a trawled plankton camera classifies ~1 million images in an hour at high accuracy, what is the new analysis that is now possible? I think the argument is that these measures are distilled/averaged/summarized to fit old analysis techniques rather than giving rise to new ones.

With respect to software pipelines in general, in other applications real-time image classification is happening, but even in-situ, where a plankton camera system needs to be more ruggedized, state of the art is real-time image classification.

As a counterexample, the 2022 paper from Bi et al "Temporal characteristics of plankton indicators in coastal waters: High-frequency data from PlanktonScope"(https://doi.org/10.1016/j.seares.2022.102283)

"The system was deployed to process in situ images collected by PlanktonScope and to provide near real time plankton density, specifically mysid shrimp data because mysid swarms could clog cooling water intake." The difference between real-time data and real-time analysis is addressed briefly in discussion section 5.2

We thank the reviewer for pointing out that our statement oversimplifies the current state of software pipelines. Recent work of Malde et al. 2020 and Bi et al. 2022 indeed shows that both hardware and software have advanced rapidly and that the bottleneck often lies in how the massive streams of classified images are aggregated, summarized and interpreted rather than in the raw classification speed itself. The text now reads as follows at line 53:

"Despite significant advances in hardware for high-throughput plankton imaging, these new instruments do not always come with a solid and easy-to-use software pipeline (Bi et al. 2022 is a rare counter-example), leaving operators with the burden of coding or adapting one themselves. Even once the data is processed, many current analysis workflows still rely on aggregating and summarizing the classified images, since the usual statistical tools used in ecology are not meant to handle such large amounts of data points. This limits our ability to leverage the full richness of these new datasets (Malde et al. 2020)."

Line 66 - this is contradictory to line 24, where CNNs of small size are sufficient. I think it is a more useful argument to mention that many CNNs take a fixed region pixel size, and plankton are highly diverse as to their relative scales as opposed to say, classifying images of jungle mammals, where there is less orders of magnitude of size difference.

Thank you for this comment. The "small size" CNN we refer to in line 24 are compact models, i.e. networks with a limited number of layers and a modest total parameter count, not networks that operate on a small input image. In practice, all images are first resized to the fixed input dimension required by the network (e.g.  $224 \times 224 \, \text{px}$ ) and padded to a square (see line 330). This preprocessing compresses the natural scale variation: larger organisms are downsized and information is condensed; while small ones are up-scaled, but interpolation cannot generate new detail, so the discriminative information that remains is limited.

Consequently, a modest-capacity CNN (few convolution layers, relatively few parameters) can already capture most of the useful information, and adding depth or increasing the number of parameters brings little extra benefit. We therefore keep the paragraph at line 60 unchanged, as it accurately describes challenges of plankton image classification. In addition, we have inserted a sentence at line 334 to emphasize the preprocessing effect aspect:

"Since all images are resized and padded to a common pixel grid, the large natural size variation of plankton is compressed, limiting the amount of scale-specific detail that can be exploited by the CNN."

Furthermore, to make clear that "small-size" refers to the architecture of the network rather than to the input image size, we have revised the sentence at line 24 as follows:

"Our findings suggest that compact CNN (i.e. modest number of convolutional layers and consequently relatively few total parameters) are sufficient to extract relevant information to classify small grayscale plankton images."

Line 80 - "It also means that no universal set of features can be produced to identify all plankton traits across instruments" - I disagree, I think that the importance of feature selection is overstated: it reduces computational overhead but costs accuracy. Most pre-CNN algorithms, when properly trained, will learn to weight the redundant or noisy features

closes to zero. Also, I think whether or not a universal set of hand-crafted features can exist is moot, because, as put in Irisson et al. 2022 (https://doi.org/10.1146/annurev-marine-041921-013023) """This really is the main progress that CNNs bring: One can forgo the considerable domain expertise required to craft appropriate features, use a pretrained feature extractor, and get results that are equally good, if not better."""

We thank the reviewer for this comment. We acknowledge that this statement can be interpreted as overstating the importance of handcrafted feature selection, especially in light of the advantages of deep-learning approaches that we discuss later in the paper. We have revised the paragraph at line 97 accordingly:

"Because handcrafted features are designed for a particular imaging system, a single universal set that works across all instruments is difficult to define; the optimal set of features tends to be instrument and dataset dependent (Orenstein et al. 2022). One solution would be to define a very large, universal feature set and leave it to the classifier to select the relevant ones for each task. But this would be a challenging task, as it requires both expertise in biology, for many taxa (to know what to extract), and in computer science (to know how to do it); feature engineering has therefore emerged as a complete research field (Guyon and Elisseeff, 2003). In the following, we will refer to these two-step methods (1 – handcrafted feature extraction and 2 – classification) as "classic approaches", in contrast to the "deep approaches" introduced later, which bypass manual feature design by training feature extractors that automatically learn relevant features for the task at hand (Irisson et al., 2022)."

Line 255 - Ellen et al. 2019 specifically uses a narrow distribution of values for a non-homogeneous background.

We thank the reviewer for pointing this out. We have removed the Ellen et al. (2019) citation, as our padding uses a homogeneous background and the reference was not applicable.

Section 2 - very strong.

Thank you!

Line 279 - Any indication that 10/5 epochs were sufficient?

We thank the reviewer for this comment. As noted at line 361, we monitored training progress by computing the loss on the validation after each epoch. The loss curves flattened before the final epoch, indicating that additional epochs would not improve performance. To illustrate this, we have added a new figure to the Supplementary Material (Figure S1) that plots the validation-loss trajectories for MobileNet V2 (10 epochs) and EfficientNet S (5 epochs) on the UVP6 dataset, confirming that the chosen numbers of epochs were indeed sufficient.

Fig 1 - Beautiful figure.

Thank you very much for this positive feedback!

Table 4 - The color scale is hard for me to discriminate between values <5% apart.

Thank you for raising this issue. The editor informed us that coloured cells cannot be retained in the final version, so these have been removed.

Line 324 - Why choose a random classifier as the baseline? Shouldn't the baseline be the classifier that always chooses the dominant class? In the provided example, 90% seems like a more appropriate benchmark to beat than 81%.

We thank the reviewer for this insightful question. A random classifier that draws labels according to the empirical class frequencies reflects the distribution of the training data while still producing a full confusion matrix. This allows us to compute class-sensitive metrics such as balanced accuracy, precision and recall, which are essential for evaluating performance on highly imbalanced plankton datasets. In contrast, a deterministic "always-predict-the-dominant-class" baseline yields an inflated accuracy (e.g. 90 % in the 11-class example) but provides no information about the many minority taxa and does not permit the calculation of those metrics. Therefore, we retained the random-class classifier as the sole baseline. The text at line 408 has been updated to emphasise this aspect:

"In datasets with strong class imbalance – such as many plankton datasets – accuracy alone can be misleading. For instance, in an 11-class dataset with one dominant class comprising 90% of the data (and each of the other classes making up only 1%), a classifier that always predicts the dominant class would achieve 90% accuracy but would provide no insight into the ten minority classes. A random classifier that draws labels according to the empirical class distribution would yield a lower-bound 81% accuracy  $(0.9^2 + 10 \times 0.01^2)$ . This baseline reflects the underlying distribution while still producing a full confusion matrix that can be used to compute metrics such as precision and recall."

**Line 361 - +15.1% seems better than "only slightly"**

Thank you for raising this issue, indeed the wording "only slightly" is not appropriate to describe a +15.1% average improvement. However, this number is largely driven by the UVP6 dataset (+42.6 and 43.8%). The text has been revised accordingly at line 463:

"In terms of overall accuracy, the CNN only showed a modest improvement on five datasets compared with the classical approach of using handcrafted features and an RF classifier (+3.5% to +13.8%) (Fig. 2). The exception was the UVP6 dataset, where the improvement was more pronounced (>40%)."

Line 500 - Another way to improve accuracy is more training data, and there is a result that shows that using plankton images from different instruments in conjunction with

"In summary, our recommendations for training a CNN to classify plankton images begin with assembling as many annotated plankton images as possible, even if images are from seemingly disparate sources."

(Ellen and Ohman 2024 - doi: 10.1002/lom3.10648)

We thank the reviewer for this comment. We agree that increasing the amount of annotated plankton images, especially when they are gathered from different instruments, can improve CNN accuracy. Enriching the information content of each image is an independent avenue for improvement. Accordingly, we have revised the manuscript at line 621 to distinguish clearly between "more data" and "richer data" and we have added the Ellen & Ohman 2024 reference:

"Therefore, improvements in classification accuracy are more likely to come from richer inputs than from larger network architectures. One way to achieve this is by increasing the quantity of annotated plankton images; pooling data from multiple instruments and sampling conditions has been shown to improve CNN accuracy (Ellen and Ohman, 2024) and is the first step towards building a so-called foundation model for plankton images. A second, independent route is to enhance the informational content of each image. For example, color cameras such as those used in the planktoscope (Pollina et al. 2022) or the Scripps Plankton Camera (Orenstein et al. 2020b), should capture more information by using multiple channels. Beyond color, additional fluorescence channels can be obtained using environmental high content fluorescence microscopy, enriching the information content of images (Colin et al. 2017); but this method can only be applied ex situ. Expanding the amount of training data and capturing richer image information should both yield gains in classification performance, albeit at the cost of greater storage and processing requirements."

Line 529 - I do not see that conclusion about geometric metadata features in Ellen et al. 2019: """Combining each individually with the geometric metadata provides a boost in performance. One possible explanation for this result is that the geometric metadata includes information about the original region of interest size, which on its own did not prove valuable, but size given a depth or temperature may have discriminative value. """

We thank the reviewer for pointing out this issue. In Ellen et al. (2019) the authors indeed reported that combining geometric metadata with geotemporal and/or hydrographic metadata did improve performance compared to geotemporal and hydrographic metadata alone, whereas geometric metadata alone did not (see Figure 13). Our original wording incorrectly implied that geometric features were never beneficial. We have therefore corrected the paragraph at line 671:

"Ellen et al. (2019) evaluated the effect of concatenating different types of "metadata" (geometric, geotemporal and hydrographic) to fully connected layers: geometric features alone did not improve model performance, whereas geotemporal and hydrographic metadata each yielded a noticeable boost, and adding geometric metadata on top of those provided an additional improvement."

Line 557 - Space after period is missing. "set.Since" Thank you for noticing this. The text has been corrected.

Line 578 - I thought the wording of the 4 questions in lines 160-164 was particularly well done. I was disappointed to not find a corresponding set of concisely worded conclusions.

Thank you for raising this point. We agree that such wording would be beneficial to the reader. The text has been rephrased accordingly at line 721:

"In summary, a small CNN achieved strong performance at plankton image classification across six realistic plankton image datasets, while being easy to apply. It unsurprisingly outperformed the classical approach of extracting a small number of handcrafted features and using a RF classifier, particularly for rare classes. Applying per-class weighting improved the detection of underrepresented classes. Surprisingly, using a large CNN did not lead to better classification performance than a much smaller one and deep features could be quite heavily compressed without loss of performance. This is likely related to the fact that plankton images, which are typically small and grayscale, are poor in informative content for CNN. Finally, the nature of the features dominated the outcome: deep features drove the performance gains, while the choice of classifier had little impact. Overall, these findings suggest that larger and more diverse training sets and/or advances in imaging hardware, rather than ever larger models, will be key to further improving plankton classification. Furthermore, metrics that emphasize the classes of interest — often the minority classes in plankton datasets — should be prioritized."

---

## Author Response (AR2)

**Responses to reviewers**

We thank the reviewer for their additional comments on the revised manuscript. Below we respond to each point individually, outlining the changes made to the manuscript (line numbers refer to lines in the manuscript with tracked changes).

Thank you for considering the suggestions. The manuscript has improved and most of my suggestions were noted or reasonably justified. There were still a few points that, in my opinion, were insufficiently addressed or answered, perhaps due to misunderstandings. So, still a few comments for the authors to consider.

My comment on acknowledging also the smaller efforts of publishing open, manually annotated datasets was addressed as follows "Earlier plankton image datasets were modest in size, typically containing a dozen or a few dozen of classes (Benfield et al., 2007), but were crucial for establishing the first classification methods." Based on this, I must assume that the authors have misunderstood my point. I did not mean earlier, first works, but all different efforts there are to expand the publicly available image libraries for reproducible and open science, as well as for the use of other users of the instruments. I know it is an issue that, within the wide field of publications on plankton recognition, it is hard to compare results between them, especially when the datasets have not been published. It is also hard even if the datasets are published, but if all the different methods are tested on different datasets. Therefore, benchmark datasets are valuable. However, to promote open science and also advance the development of models with a diverse training set, it is also important to promote smaller efforts in publishing open datasets for the purpose of model development. For future avenues of developing automated recognition of plankton, the more diverse training datasets we have available from multiple sources, the better. I believe this has also been improving a lot in recent years. Therefore, it would be nice to mention that, besides the "three major plankton image datasets" used in this study, there is an expanding effort to publish manually annotated datasets.

To make it easier and get some idea of how much effort there has been on this, I compiled a list (surely not comprehensive) for you to have a look at.

-European IFCB users have gathered links to open datasets
https://nordicmicroalgae.org/annotated-images/
-Table 2 in Eerola et al. 2024: listed published datasets of plankton that were related to publications.
-Table 1 in Kareinen et al. 2025 (Self-Supervised Pretraining for Fine-Grained Plankton Recognition https://arxiv.org/html/2503.11341v2) contains some more recently published datasets.
-A new version of the ZooLake dataset https://doi.org/10.25678/000C6M
-A FlowCam dataset https://doi.org/10.5281/zenodo.16679297
-A Flowcam dataset https://zenodo.org/records/16840846

We thank the reviewer for raising this issue, the we have inserted the following paragraph at line 154 to highlight the importance of publishing smaller open datasets:

*Beyond publishing large reference datasets, as we strive to do in this work, another avenue for progress is the collection of many diverse, albeit smaller, datasets. This is typically the first step for the creation of "universal" foundation-type models. The push towards more open and reproducible science has helped in this respect and several local datasets have been published: e.g. Table 1 in Kareinen 2025, Table 2 in Eerola et al. 2024.*

My previous comment: "Figure 2: Why did you choose to show accuracy and not F1 score in the first panel (the same comment also goes for the subsequent figures)? What is the Random classifier? It was mentioned in a paragraph starting from line 350, but it would require a better explanation." -I would still argue on behalf of adding F1 scores. I don't think that it would be redundant, as it is very difficult to go and look at the F1 scores from the tables. I think that if you want to highlight the fact that the naive baseline can be misleading, you could add the F1 on top of the accuracy bars as dots. The F1 score is a very common metric to report, and it would make sense to have it in the figures as well to make comparisons easier. It would also be interesting to show how different picture F1 gives versus balanced accuracy, thus not being a redundant addition to the first panel in figures 2, 3, and 4.

We appreciate the suggestion to display F1 scores alongside accuracy-type metrics. To evaluate their added value, we generated supplementary plots that show micro-F1 versus accuracy and macro-F1 versus balanced accuracy for every model (Figure S2, presented below).

[Figure]

*Figure S2: Relationship between F1 and accuracy-type metrics. (a) Micro-averaged F1 against accuracy and (b) macro-averaged F1 against balanced accuracy; for every model across all datasets. The dotted diagonal represents the 1:1 line, highlighting where the two metrics convey identical information.*

The plots reveal two key observations that support our decision to retain accuracy and balanced accuracy as the primary figures:

1. Micro-F1 vs. accuracy: all models fall on the 1:1 line, indicating that micro-F1 conveys exactly the same information as accuracy.
2. Macro-F1 vs. balanced accuracy: again, every model lies close to the 1:1 line, confirming that macro-F1 closely resembles balanced accuracy for the class-imbalanced scenario we consider.

Because both F1 variants collapse onto the same trends as the accuracy-based measures for all models, we conclude that adding F1 bars would not improve interpretability but only hemper readability of Figures 2-4. We therefore keep accuracy and balanced accuracy in said figures, we add Figure S2 to the supplementary materials and we provide the full set of F1 values (alongside with other metrics) in the supplementary Table S8 for readers who wish to inspect them. We think that this decision preserves clarity in the main figures while still offering complete metrics for full transparency. The following edits were made to the manuscript.

Paragraph updated at line 336:
*"Usual metrics were computed: accuracy score (percentage of objects correctly classified), balanced accuracy, macro-averaged F1-score, micro-averaged F1-score, class-wise precision (percentage correct in the predicted class) and recall (percentage correct within the true class)."*

Paragraph updated at line 352:
*"To focus on these classes, we also computed the average of precision and recall per class, weighted by the number of objects in the class, but using only plankton classes, i.e. the target classes (Owen et al., 2025)."*

Inserted in legends of figures 2, 3 and 4:
*"All values, including F1-scores, are reported in Table S8."*

Inserted at line 391:
*"The same applies for F1-scores: macro-F1 captures the failure of the random classifiers, while micro-F1 mirrors accuracy (Fig. S2)."*

My previous comment: "375-385: Wouldn't it be important to find a harmonic mean between precision and recall rather than emphasize the importance of precision and detection of rare classes over recall?" -Yes, but my point was that you chose to present results only of the weighted models from here onwards, which had lower precision but higher recall (it seems I wrote them the other way round in my comment). The question was why you didn't choose

the model based on the balance between these two, but chose the strategy highlighting recall. I know that if you want to get better recall for the plankton classes, you chose the weighted model, but I would not want my classes to be disturbed by many false positives either. This is why I asked why not choose a strategy highlighting the harmonic mean between these two instead of choosing a strategy that highlights getting correct hits of the rarer classes, but with the consequence of getting false positives? This is also a topic: with a closed set classification, one needs to choose which strategy to follow; with open set classification methods, the target is to accurately identify images belonging to the existing classes, but also to identify or filter out the ones that don't. I think this is a topic that should be raised in the section on costs and benefits of using CNNs, a limitation of closed-set classification systems, and also an alternative approach of filtering out images of too low prediction scores (thresholding), which is, of course, also nonideal.

We thank the reviewer for this valuable comment. We agree that favouring recall is not the only possible strategy; it is the one we chose in this study because it aligns with common goals when training such models. We have inserted the following paragraph in the discussion at line 601 to discuss this trade-off:
*"Weighting improves the recall of rare classes but reduces their precision, reflecting the classic precision–recall trade-off. When downstream analysis involves manual verification, higher recall is advantageous because a few false positives in rare classes can easily be corrected while missed detections would likely be lost among the most numerous classes and not easily recovered. Conversely, in high-throughput monitoring through imaging, where human review of all samples is infeasible, emphasizing precision reduces spurious detections at the cost of under-estimating true abundances. In such settings, post-hoc confidence thresholding (e.g. Faillettaz et al., 2016; Luo et al., 2018) offers a pragmatic compromise, albeit an imperfect one. In all situations, using various intensities of class weighting is a flexible solution to adapt the classifier to the study's objective."*